# Geometric and mechanical guidance: Role of stigmatic epidermis in early pollen tube pathfinding in arabidopsis

Lucie Riglet[1¤], Catherine Quilliet[2], Christophe Godin[1], Karin John [ID][2*], Isabelle Fobis-Loisy [ID][1*]

**1** Laboratoire Reproduction et Développement des Plantes, Univ Lyon, ENS de Lyon, UCB Lyon1, CNRS, INRAE, INRIA, Lyon, France, **2** Université Grenoble-Alpes, CNRS, Laboratoire Interdisciplinaire de Physique, Grenoble, France

¤ Current address: Sainsbury Laboratory, University of Cambridge, Cambridge, United Kingdom
* karin.john@univ-grenoble-alpes.fr (KJ); isabelle.fobis-loisy@ens-lyon.fr (IFL)

**Data availability statement:** All relevant data are within the manuscript and its Supporting information files.

## Abstract

In *Arabidopsis thaliana*, successful fertilisation relies on the precise guidance of the pollen tube as it navigates through the female tissues to deliver sperm cells to ovules. While prior research has focused on pistil signals directing pollen tubes towards the ovules, the pollen tube growth within the stigmatic epidermis has received limited attention. Our recent work comparing wild-type pollen tube paths on wild-type and *katanin1-5* stigmatic cells, revealed a tight connection between pollen tube directionality and mechanical properties of the invaded stigmatic cell. Given that most mechanical properties of the stigmatic tissue are experimentally challenging to access, we used mathematical modelling to investigate the mechanisms underlying early pollen tube guidance through the papilla cell wall. We found that in *ktn1-5*, the wild-type pollen tube navigates freely across the curved papilla surface, following curves close to geodesics, whereas the wild-type papilla imposes directional guidance. The order of magnitude analysis of the mechanical forces required for pollen tubes to progress at the papilla surface indicates that both the elongated geometry of the papilla and the difference in rigidity of its cell wall layers combine to efficiently orient the pollen tube towards the papilla base.

## Author summary

Understanding how pollen tubes navigate within the pistil to reach their targets is a fundamental question in plant reproduction. This process begins with the intricate cell-to-cell communication initiated by the contact between the pollen grain (the carrier of male sperm cells) and the stigmatic papillae (receptive platform of the female reproductive organ). Employing a multidisciplinary approach combining methods and

**Funding:** This work was supported by "La Mission pour les initiatives transverses et interdisciplinaires (MITI), defi Mécanobiologie 2018" (https://miti.cnrs.fr/appel-a-projets/mecanobiologie), from the Centre National de la Recherche Scientifique (to CQ, KJ and IFL) and by the "Institut Rhonalpin des Systemes Complexes" (to CQ, KJ and IFL; http://www.ixxi.fr). The funders had no role in study design, data collection and analysis, decision to publish, or preparation of the manuscript.

**Competing interests:** The authors have declared that no competing interests exist.

concepts from biology, geometry and mechanics, we explored how the growth direction of pollen tubes is influenced by the biological and physical properties of the papilla cell they invade. By comparing model predictions with experimental data, we identified a potential guidance mechanism driven by the quasi-cylindrical geometry of the papilla and the elastic properties of its cell wall.

## Introduction

In the flowering plant *Arabidopsis thaliana*, reproduction initiates upon the arrival of a pollen grain at the receptive surface of the female reproductive organ, ~~also~~ called stigma (Fig 1A and 1B). Once landed, the pollen grain germinates, forming a pollen tube responsible for transporting sperm cells to ovules deeply embedded within the pistil (Fig 1C) [1]. Precise guidance of the pollen tube is essential to ensure its correct path through the female tissues, preventing misrouting and securing the delivery of male gametes. While prior research has focused on pistil-produced chemical, electrical and mechanical signals directing pollen tubes towards the ovules [1–3], the early orientation of pollen tubes within the stigmatic epidermis has received limited attention. In *A. thaliana*, pollen tubes first penetrate the cell wall of the stigmatic cells (or papillae) and then grow toward the stigma base, while being constrained within the two layers of the rigid stigmatic cell wall (Fig 1D and 1E) [4–6].

We previously published that the microtubule-severing enzyme KATANIN (KTN), by acting both on cortical microtubule (CMT) and cellulose microfibril (CMF) organization, conferred particular mechanical properties to the papilla cell wall, correlating with the guidance of pollen tubes [6]. On *ktn1-5* papillae, where the cell wall displayed impaired mechanical properties (i.e. isotropic CMT and CMF arrays, softer cell wall), wild-type (WT) pollen tubes exhibit a helical growth pattern (coiled path), occasionally growing in the opposite direction of the stigma base and ovules. In contrast, on WT papillae, characterized by anisotropic CMT and CMF arrays and stiffer cell wall, the WT pollen tubes maintain a relatively straight trajectory towards the stigma base (Fig 1F and 1G). These correlations between the mechanical properties of the papillae cell wall and paths of the pollen tubes suggest that mechanical cues could serve as guidance mechanisms at the earliest step of the male - female interactions.

Due to the technical challenges associated with a full characterization of the pollen tube growth in a complex environment (e.g. molecular details of the tip growth, wall stresses and turgor pressure of the papilla; [7]), we used, here, phenomenological computational modelling to investigate the mechanisms by which the papillae influence the growth of the pollen tubes. In this approach the pollen tube is characterised by a bending rigidity which provides resistance against changes in its growth direction due to external guidance cues. We tested our simulation by comparing the model predictions to experimental data from our prior observations [6] as well as new experimental data concerning both WT and *ktn1-5* papillae. Our main findings showed that the coiled paths of WT pollen tubes on *ktn1-5* papillae stem from the absence of guidance, thus corresponding to curves close to geodesic trajectories on the curved papilla surface. In contrast, the straight growth pattern observed on WT papillae requires a guidance cue that induces the pollen tube tip to redirect its growth along the papilla longitudinal axis.

By combining experimental data with simulations, we propose a mechanism wherein the guidance of the pollen tube is driven by the quasi-cylindrical geometry of the papilla and the elasticity of its cell wall. We tested, two, mutually not exclusive, elasticity effects that may contribute to growth guidance: (i) a rigidity contrast between the outer and inner leaflets

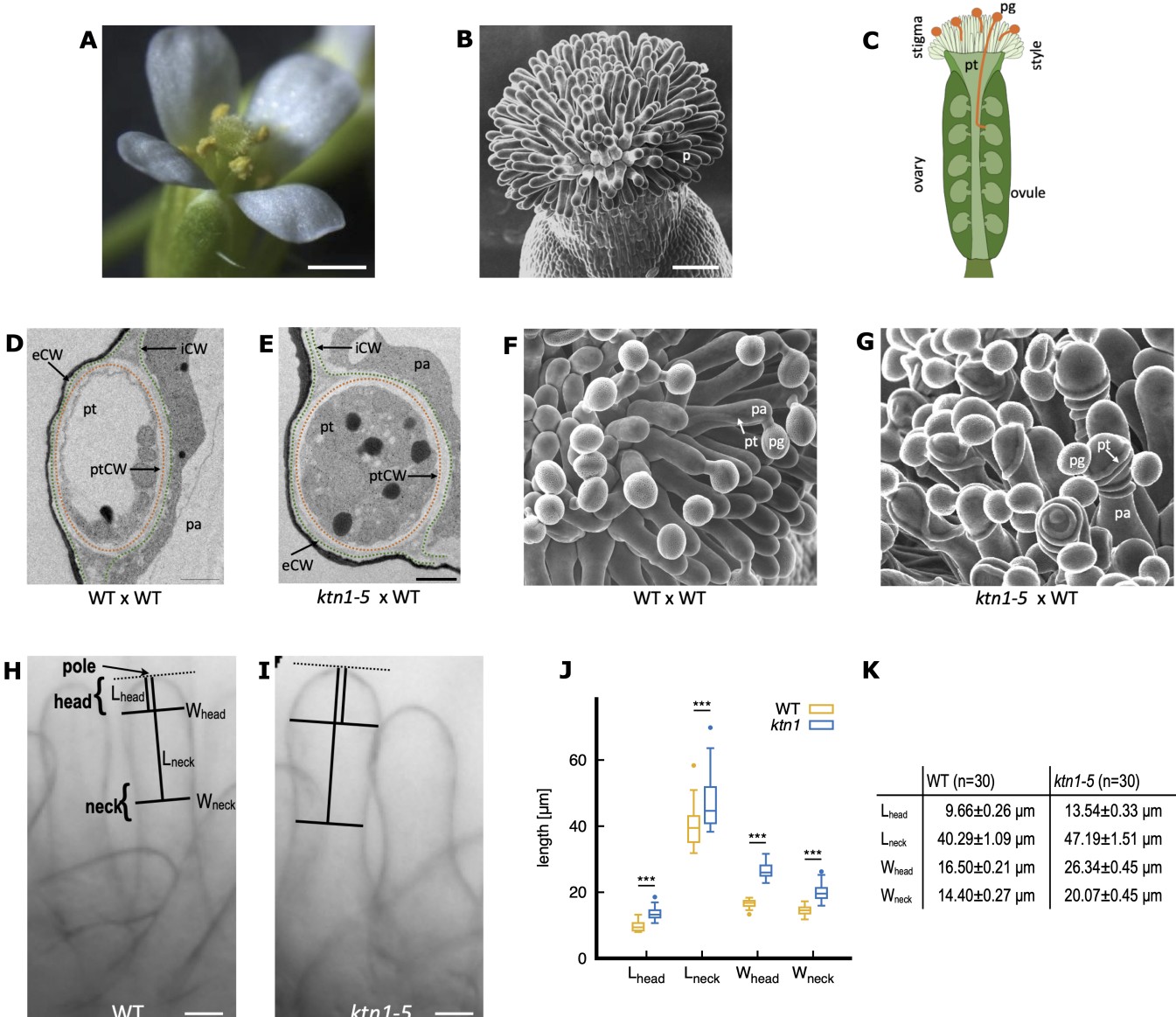

**Fig 1. Pollen-stigma interactions in *A. thaliana* and characterisation of papilla shape. (A)** A mature *A. thaliana* flower. The pistil at the center of the flower is surrounded by six anthers containing the male gametophytes (yellow pollen grains). Scale bar = 500 $\mu$m. **(B)** View of the top part of the pistil, the stigmatic epidermis, imaged by scanning electron microscopy (SEMi), and composed of hundreds of elongated papillae (p). Scale bar = 100 $\mu$m. **(C)** Schematic representation of the pollen tube journey within the pistil tissue. A pollen grain (pg) released from the anthers lands on a papilla and germinates a pollen tube (pt) which transport the male gametes through the stigma, style and ovary, towards the ovules for fertilization. **(D,E)** Transversal section of a WT (D) and *ktn1-5* (E) papilla pollinated with WT pollen and observed by transmission electron microscopy. The cuticle appears as an electron-dense black layer. The pollen tube progresses within the stigmatic cell wall, between its internal (iCW) and external (eCW) layers. For better visualisation, stigmatic (iCW + eCW) and pollen tube (ptCW) cell walls are highlighted with a green and orange dashed line, respectively. The original information was previously published in [6]. Images displayed here differ from the ones in [6]. Scale bar = 1 $\mu$m. **(F, G)** SEMi images of WT (F) and *ktn1-5* (G) papillae pollinated with WT pollen grains. Most of the pollen tubes go straight towards the base of the WT stigma (F) whereas pollen tubes make loops in the *ktn1-5* papillae (G). The original information was previously published in [6]. Images displayed here differ from the ones in [6]. Scale bar = 10 $\mu$m. **(H,I)** Light microscopy images of WT (H) and *ktn1-5* papillae (I) with relevant shape descriptors $L_{head}$, $L_{neck}$, $W_{head}$ and $W_{neck}$. Scale bar = 10 $\mu$m. **(J)** Box plots of the shape descriptor dimensions measured on 30 papillae from 3 WT or 6 *ktn1-5* stigmas. The horizontal bar in the boxes corresponds to the mean value. Statistical analysis was based on a non-parametric Wilcoxon Rank Sum test. *** indicates a p-value <0.001. The label *ktn1* refers to the *ktn1-5* mutant. **(K)** Means of the shape descriptor dimensions +/- standard error of the mean. Measurements are provided in S2 Table. n denotes number of papillae analysed.

of the papilla wall and (ii) a direction-dependent mechanical anisotropy within the papilla wall. Our simulations provide strong support for the first hypothesis and effectively explain how coiled and straight pollen tube growth arises, successfully reproducing our experimental observations.

## Results

For clarity and ease of comprehension, the major papilla and pollen parameters used in this study, along with their corresponding biological/physical interpretations are listed in S1 Table.

### Shape differences in WT and *ktn1-5* papillae

Both WT and *ktn1-5* papillae share an overall quasi-cylindrical shape resembling a bowling pin (Fig 1F and 1G). In our prior study [6], where we measured the total length of the papillae and the width of its head, we observed some variations between genotypes. Here, to provide a more detailed characterisation of the WT and *ktn1-5* papilla morphology, we quantified more accurately their shapes using four geometric shape descriptors (Fig 1H and 1I).

Papillae display a nearly hemispherical head region, characterized by measurements $L_{head}$ and $W_{head}$ (head length and width, respectively). This head region gradually transitions into a cylindrical shaft region that narrows at the neck, characterized by the distances $L_{neck}$ (from the pole to the neck) and $W_{neck}$ (neck width). We found significant differences (Fig 1J and 1K and S2 Table), with WT papilla cells being more slender with a narrower head ($W_{head} = 16.50\,\mu m \pm 0.21\,\mu m$) and neck ($W_{neck} = 14.40\,\mu m \pm 0.27\mu m$) than *ktn1-5* counterparts ($W_{head} = 26.34\,\mu m \pm 0.45\,\mu m$; $W_{neck} = 20.07\,\mu m \pm 0.45\,\mu m$).

Next, we wondered whether these differences in papilla morphology could impact pollen tube trajectories.

### Geodesics as reference trajectories on a pin-like surface

Using a geometrical model, we first investigated how variations in shape of a pin-like structure, specifically constriction in the neck region, could influence the theoretical trajectories of an object moving along its surface.

In the absence of any forces, other than those that keep it on the surface, an object advancing locally straight on a curved surface follows a curve on this surface called a geodesic [8]. Geodesics correspond to the length-minimizing curves between pairs of points on a (smooth) surface, and possess the remarkable property that, at any given point on the surface and in any specified direction, a single, unique geodesic passes through that point in that direction [9].

To get a first insight into the role of the neck constriction in altering geodesic trajectories on nearly cylindrical surfaces, we computed geodesic paths with fixed initial positions and directions on pin-like shapes with varying neck diameters, as illustrated in Fig 2. As the neck diameter decreases (D in Fig 2), the geodesic (yellow curve) changes. For instance, decreasing the neck diameter by 38% (from 0.43 to 0.27) resulted in markedly different coiling patterns (from 1 turn to 3.5 turns before reaching the bottom region), especially around the narrowest region of the pin-like shape. In extreme scenarios, where the neck became exceedingly thin, geodesics could no longer cross the neck and remained confined to the upper part of the pin-like shape (rightmost two shapes in Fig 2). The tendency of the geodesic to coil in the region of constriction is a direct consequence of Clairaut's theorem: This theorem states that on surfaces of revolution, like for example pin-like shapes, geodesic trajectories should follow the constraint $R \sin \varphi = const.$, where $R$ represents the distance of a point $P$ on the geodesic

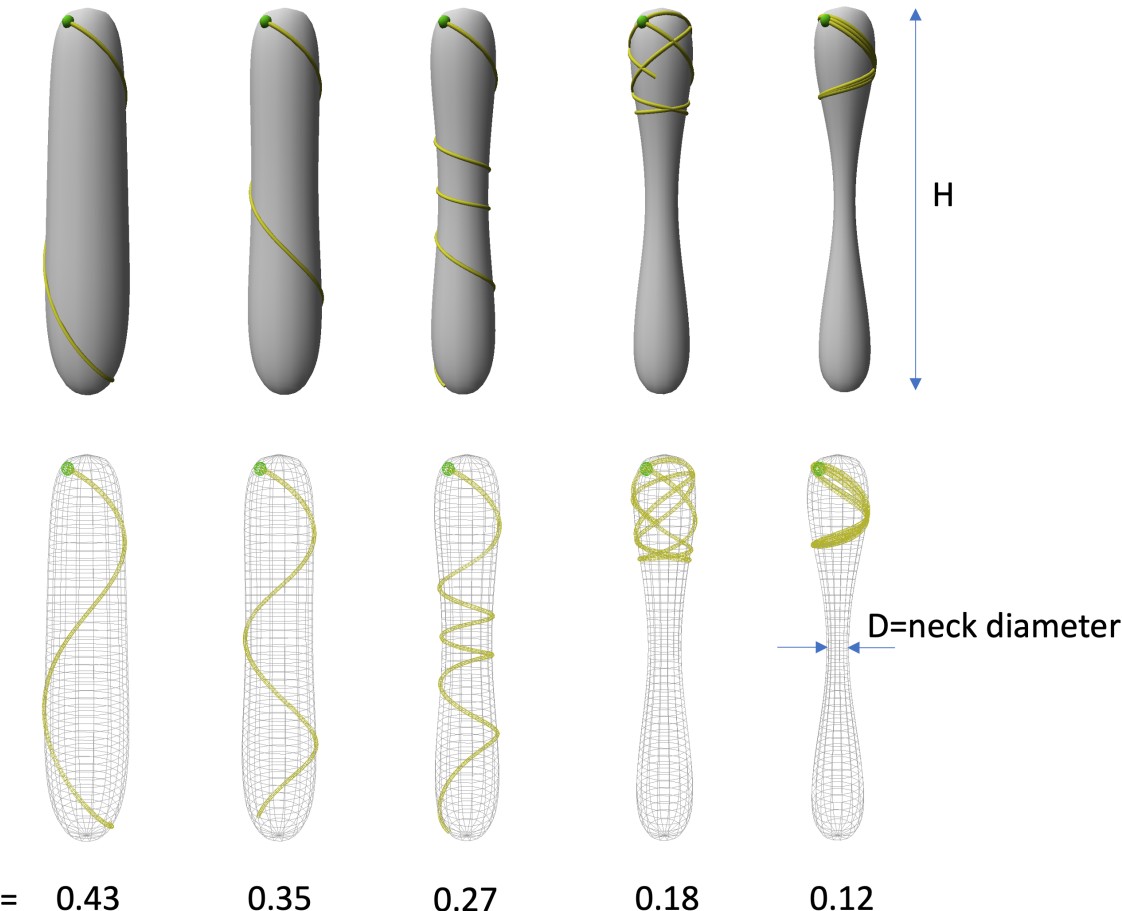

**Fig 2. Theoretical geodesic trajectories on pin-like surfaces of varying shapes.** This sequence displays pin-like structures, characterized, from left to right by a gradual decrease in neck mid-height diameter (D) while maintaining a constant height (H). The top row features 3D surface rendering of the shape. Bottom row: wire-frame rendering where the whole geodesic trajectory is visible (yellow curve). All the geodesics start at the same position (green dot) at point $P_0$ with azimuth = 0 and altitude = -0.1 with respect to the pole of the pin structure, and with an initial inclination angle of 28° downward with respect to the circumferential direction. D is given in arbitrary units (a.u.) and H=2.0 a.u. Note that the curve can cross itself (rightmost two situations).

curve to the axis of revolution of the surface **S** and $\varphi$ represents the angle between the tangent of the geodesic and the longitudinal line of **S** going through that same point $P$ [10]. As an example, on cylindrical surfaces ($R$ is constant), geodesic lines correspond to helices with a constant pitch and slope. For our nearly cylindrical surfaces, in regions where the radius $R$ of the pin structure decreases, the trajectory tends to orient more in the circumferential direction, i.e. $\sin \varphi$ increases to maintain a constant product $R \sin \varphi = const.$, and coil more. This property is further discussed and illustrated in Ref [11] and in the references therein.

Thus, according to this theoretical model, where the tube is assumed to grow as straight as possible of the surface, without mechanical or chemical cues, the presence of constrictions in the neck region enhances the coiling behaviour of geodesic trajectories on nearly cylindrical shapes. However, it is important to notice that substantial alteration in the neck diameter are required to induce drastic qualitative modifications of the trajectories as illustrated in Fig 2.

Interestingly, the pollen tube trajectories on WT papilla (Fig 1F), despite starting in various direction, do not exhibit helical paths in the cylindrical papilla shaft nor an increased coiling behaviour in the neck region which suggests that they do not follow geodesics. In contrast, the coiling behaviour of the pollen tube trajectories on *ktn1-5* papillae (Fig 1G), is reminiscent of that of geodesic lines on pin-like surfaces. To systematically explore the influence of papilla shape and possible guidance cues in pollen tube growth, we developed a minimal phenomenological model that integrates quantification of the WT and *ktn1-5* papillae geometry, along with well-known features of pollen tube growth.

## A minimal phenomenological mechanical model of pollen tube growth on the papilla surface

Using the previously defined papilla descriptors (Fig 1H and 1I), we fitted mathematical functions to derive analytical three-dimensional virtual WT and *ktn1-5* papillae surfaces **S** (Fig 3A). Thereby, papilla shapes are parameterized using cylindrical coordinates $[\theta, z]$, where $\theta$ and $z$ respectively represent the angular direction and the distance along the papilla axis (measured from the papilla pole).

Numerous observations from our previous studies [6,12] consistently support the finding that pollen grains tend to attach to the head region of the papilla (see as examples Fig 1F and 1G and S1 Fig). Attachment to the neck region or even below is rare and might be geometrically constrained, i.e. the pollen grain does not fit in between papillae. As a result, in our model, pollen grain attachment site and the corresponding landing position $z_0$ (Fig 3B) are constrained to distances $z_0 < 2L_{head}$ ($L_{head}$ as indicated in Fig 1H). In addition, we assumed that pollen grains attach with equal probability on any point of the papilla surface provided that $z \leq 2L_{head}$ (For an alternative hypothesis where pollen grains attach preferentially to the papilla pole see Materials and Methods). After emerging from the grain, the pollen tube outgrowth has an initial direction, represented by the $\varphi_0$ angle made with the longitudinal papilla axis (Fig 3B). We assumed that this initial direction is devoid of any directional bias, implying that all potential initial directions are equally probable. By convention, $\varphi_0 = 0$ ($\varphi_0 = 180°$) indicates that the tube grows along the long papilla axis towards its base (respectively pole), while an angle of $\varphi_0 = 90°$ indicates an initial trajectory oriented along the papilla circumferential direction.

The pollen tube, a tip-growing cell, undergoes unidirectional elongation, with its expansion restricted to the very tip of the cellular protrusion [13]. Moreover, in *A. thaliana*, the pollen tube progresses within the papilla cell wall, confined between the two leaflets of the cell wall (Fig 1D and 1E and Ref [12]), keeping it on the papilla surface (i.e. preventing penetration into the papilla volume or escape into the air). A precise description of the known molecular processes and mechanics governing pollen tube growth within the papilla cell wall (see e.g. Refs [7,14,15]) would require a numerically costly multi-scale approach with many unknown parameters. Here, in our phenomenological model, we considered the pollen tube as an inextensible filament with bending rigidity $\chi$ whose trajectory is maintained on the papilla surface **S**. This bending rigidity serves as a resistance against changes in the growth direction at the pollen tube tip, consistent with *in vitro* experimental observation that pollen tubes grow straight over several hundreds of micrometres in the absence of mechanical forces or chemical guiding cues [14–16].

The growth direction of the tip is obtained from the equation of momentum conservation at the filament tip, which may include an external force of mechanical and/or chemical origin. In detail, the pollen tube path is given by $\mathbf{X}(s)$ with $s$ denoting the length of the trajectory from a given initial position. Denoting $L(t)$ the length of the pollen tube at time

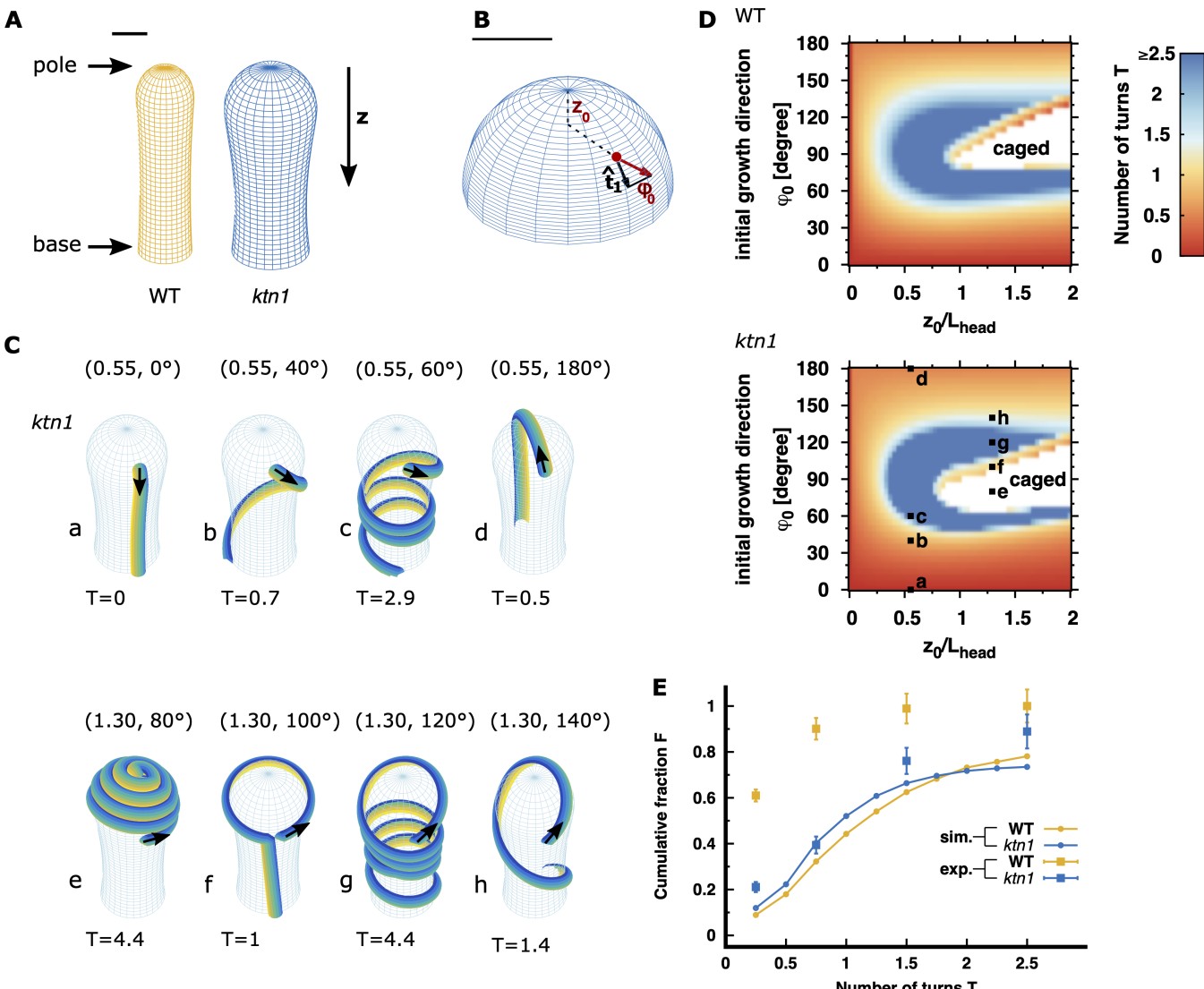

**Fig 3. Papilla prototypes and pollen tube trajectories without guidance on WT and _ktn1-5_ papillae. (A)** Three dimensional shapes of WT and _ktn1-5_ papillae. The papilla long axis is oriented along the _z_ axis. _z_ = 0 corresponds to the papilla pole. Scale bar = 10 $\mu$m. **(B)** Three-dimensional view of the top region of a _ktn1-5_ papilla showing the initial position of the pollen grain ($z_0$, red dot) and the initial direction of the pollen tube upon emergence from the grain ($\phi_0$, red arrow). The vector $\hat{t}_1$ represents a papilla surface tangent vector pointing in the longitudinal direction. 0° (180°) indicates an initial tube direction towards the papilla base (pole). Scale bar = 10 $\mu$m. **(C)** Examples of simulated pollen tube trajectories on _ktn1-5_ papilla surfaces, varying in initial positions $z_0$ of the pollen grain and initial directions $\varphi_0$ of the emerging pollen tube (indicated by black arrows). The numbers in brackets denote the normalized initial positions ($z_0/L_{head}$) and the initial directions ($\varphi_0$) for each trajectory. T (below each papilla) represents the number of turns the pollen tube makes to reach the papilla base . Each configuration is labelled from a to h. Trajectories on WT stigma with identical initial conditions are shown in S2 Fig. **(D)** Morphological phase diagrams of the pollen tube turn number T depending on the initial pollen grain position $z_0$ (normalised to the papilla distance head length $L_{head}$) and the initial pollen tube direction ($\varphi_0$) on WT and _ktn1-5_ papillae. $z_0/L_{head}$ = 0 denotes the papilla pole, $z_0/L_{head}$ = 1 the frontier between the head and cylindrical shaft, and $z_0/L_{head}$ = 2 the pollen grain landing limit. The colour code indicates the number of turns the trajectories undergoes before it reaches the papilla base. In the white region (caged), the tube path is trapped by its own trajectory, preventing it from reaching the papilla base; such trajectories were categorised as having _T_>2.5. The letters a to h correspond to the example configurations depicted in (C). **(E)** Comparison of simulated (solid lines) and experimental (squares) cumulative distributions of pollen tube turn numbers. To calculate the cumulative fraction for experimental data, we utilized data from Ref [6], where we examined 251 WT and 327 _ktn1-5_ pollinated papillae; error bars represent the standard error of the mean. The label _ktn1_ refers to the _ktn1-5_ mutant.

$t$, the position of the elongating tip $\mathbf{P}(t)$ is given by the position of the filament extremity $\mathbf{P}(t) = \mathbf{X}(s = L(t))$. We assume that over a small time lapse $\delta t$, the tip grows by a length $\delta s$ in a direction $\hat{\mathbf{t}}(L)$, which results in a new tip position $\mathbf{P}(t + \delta t) = \mathbf{X}(L(t + \delta t)) = \mathbf{X}(L(t) + \delta s)$. To calculate the tip growth direction $\hat{\mathbf{t}}(L)$, the tube bending $\hat{\mathbf{t}}(L) - \hat{\mathbf{t}}(L - \delta s)$ is decomposed into two components: (i) the bending in the plane defined by the surface normal and the tangent to the trajectory at the tip (i.e. to keep the tip trajectory on the papilla surface), and (ii) a bending either to the left or to the right in the local plane tangent to the papilla surface. We assume that this bending in the tangent plane is due to an external force (torque) that induces a rotation of the pollen tube and that comes from potential guidance cues (see Eq (13) in **Materials and Methods**). Based on our previous observations of pollinated stigma [6,12], we noted that the advancing pollen tube tip is unable to cross its own path. This property, that we termed "self-avoidance", is visually depicted in S1 Fig. In our model, self-avoidance is implemented by representing the pollen tube volume as a succession of spheres (at position $\mathbf{X}(i\delta s)$, where $i$ is an integer and $\delta s$ is an small increment along the arc length $s$ of the curve describing the tube trajectory) and cylinders (between positions $\mathbf{X}(i\delta s)$ and $\mathbf{X}(i\delta s + \delta s)$) which cannot be penetrated by the outgrowing tip. At each simulation step, we tested whether the growing tip penetrates into the excluded volume of a previously deposited pollen tube. When a potential penetration is identified, the growth direction is not determined from the equation of momentum conservation (13). Instead, we minimised the corresponding potential function (see S1 Text) ensuring that the excluded volume is not violated.

## Pollen tube growth on *ktn1-5* papillae follows curves close to geodesics modulated by the property of self-avoidance

Using the above computational model, we started by simulating pollen tube trajectories on virtual papilla in the absence of guidance cues from the stigmatic side (i.e. in absence of a torque acting on the tube tip in the local plane tangent to the papilla surface).

The behaviour of pollen tube growth was assessed by calculating the number of turns T, a methodology previously employed by [6]. Experimentally, the turn number is defined as the number of revolutions a pollen tube makes around the papilla axis down to the observable papilla base (denoted as base in Fig 3A), which corresponds to a distance of approximately $z = 60 \mu$m. We performed simulations across a broad range of initial pollen parameters $z_0$ and $\phi_0$, considering the self-avoidance property. Fig 3C provides examples of simulated tube trajectories on *ktn1-5* papilla surfaces, along with the corresponding turn numbers T. For comparison, simulated trajectories on WT papillae are shown in S2 Fig.

To integrate the entire range of configurations, we represented the turn number depending on the initial pollen parameters ($z_0$ normalized to the papilla head length $L_{head}$ and $\varphi_0$) using a morphological phase diagram (Fig 3D). Both morphological phase diagrams for WT and *ktn1-5* papillae display only slight differences, suggesting that papilla shape heterogeneity (Fig 1H–1K) does not significantly impact pollen tube paths. This confirms our conjecture based on the coiling behaviour of reference trajectories on a pin-like surface (Fig 2), where we observed that substantial differences in geometry (e.g. neck constriction) are required to induce a change in the coiling behaviour of an object moving along geodesics. Both phase diagrams show that trajectories originating from the papilla head region, particularly close to the pole with $z_0/L_{head} \leq 0.25$, consistently reach the base with relatively low turn numbers whatever the initial direction of the tube (orange colour in Fig 3D). This mainly lies in the symmetrical and spherical shape of the papilla head region, where all initial angles $\varphi_0$ result in similar types of trajectories, corresponding to longitudinal lines (geodesics) which either

go directly to the papilla base (T≈ 0) or cross over the papilla pole before going to the base (T≈ 0.5). For trajectories starting at $z_0/L_{head} \geq 0.25$, the resulting paths highly depend on the initial direction of the tube. When a tube starts with an initial direction pointing close to the circumferential direction ($50° < \varphi_0 < 130°$), it exhibits a high turn number (blue color in Fig 3D, illustrated in Fig 3C, labels c and g). In specific initial conditions, the tube tip is significantly deflected by its own path (Fig 3C, label f) or self-constraints (caged in white color in Fig 3D, illustrated in Fig 3C, label e) underscoring the importance of the self-avoidance property which can lead to completely divergent trajectories even with minor changes in the initial pollen parameters. In contrast, when a tube starts with either a low ($\varphi < 30°$) or a high ($\varphi > 150°$) initial angle, it reaches the base with a low turn numbers, either directly (T≈ 0, illustrated in Fig 3C, label a) or passing over the papilla head region (T≈ 0.5, illustrated in Fig 3C, label d) respectively.

To conduct a comprehensive global comparison between simulated and experimentally observed tube trajectories, we computed the cumulative fraction of turn numbers F(T), which represents the fraction of papillae with turn number ≤ T (Fig 3E and Materials and Methods for further details). To calculate the experimental cumulative fraction, we utilised data from [6]. We found that the cumulative fractions for simulated trajectories on both WT and *ktn1-5* virtual papillae (orange and blue solid curves in Fig 3E) closely aligns, as expected from the phase diagrams (Fig 3D). Around 40–50% of the simulated pollen trajectories completed at most one turn to reach the papilla base, and a significant proportion of tubes (about 20%) reached the papilla base exhibiting a high number of turns (exceeding 2.5). Importantly, our analysis revealed that the simulated distributions reproduce fairly well the experimental distribution of pollen tube turns on *ktn1-5* papillae (blue squares, Fig 3E) whereas significant disparities were observed when compared with the experimental trajectories pollen tubes make on WT papillae (orange squares, Fig 3E).

In summary, the parameters used in our simulations were sufficient to reproduce the strong coiling behaviour of pollen tube trajectories experimentally observed on *ktn1-5* papillae. Given that on a curved surface, when no force is acting on its tip (apart from those maintaining surface contact), a tube advances locally along the straightest possible path following geodesic trajectories (see section "Geodesics as reference trajectories on a pin-like surface"), our comparison between simulation and experiment strongly suggests that pollen tube trajectories on *ktn1-5* papillae follow geodesics or curves close to geodesics, determined entirely by their initial position ($z_0$) and initial direction ($\varphi_0$), with modulation from the self-avoidance property. In contrast, replicating pollen tube configurations on WT papillae, where pollen tubes reach the papilla basis by making ≤ 1 turn, indicates a clear deviation from geodesics, and requires additional guiding factors.

## Pollen tube growth on WT papillae requires guidance cues from the papilla side

We then explored how pollen tube in WT papillae deviate from geodesics paths, resulting in fewer turns. Considering that the pollen tube grows within the papilla cell wall constrained to follow the papilla surface, deviation from geodesic trajectories requires the presence of a force normal to the pollen tube growth direction in the plane tangent to the papilla surface. We therefore performed numerical simulations of pollen tube trajectories using the same framework as in the previous section (see also **Materials and Methods**), with the addition of an alignment force that orients the tube growth along the longitudinal axis of the papilla. This alignment force is symmetric with respect to the reorientation of the pollen tube growth direction towards the papilla base or pole and is proportional to $\sin(2\varphi)$, where $\varphi$ denotes

the angle between the direction of the advancing tip and the papilla long axis. The expression $\sin(2\varphi)$ assures that tip growth already oriented towards the papilla base ($0° \leq \varphi < 90°$) will align completely with the long axis of the papilla towards its base. For growth oriented towards the papilla pole ($90° \leq \varphi < 180°$), the growth direction will align completely with the long axis towards the papilla pole. Due to symmetry, the alignment force for growth into the circumferential direction ($\phi = 90°$) is zero. The magnitude of the alignment force (i.e. its maximum orientational force at $\phi = 45°$ and $\phi = 135°$) is determined by the adimensional constant $\mu$, which quantifies the ratio of the reorientation force to the internal resistance of the pollen tube against directional changes (provided by its bending rigidity). At this stage, it is reasonable to assume that alignment forces exclusively operate in the cylindrical part of the papilla for $z \geq L_{\text{head}}$, (Fig 4A), since the papilla head region is likely isotropic in terms of its geometry and cell wall properties due to its nearly spherical shape [17].

The impact of introducing an alignment factor $\mu > 0$ on pollen tube trajectories on WT papilla surface is depicted in Fig 4B. In configurations where pollen tubes previously exhibited significant coiling in the absence of growth guidance (Fig 4B, labels a–c), they now adopt straighter trajectories in the presence of a guiding cue (Fig 4B, labels d–f), reaching the papilla base with minimal turning T ≤ 0.5. Notably, a guidance strength of $\mu = 0.1$ is sufficient to direct pollen tubes towards the papilla base with few turns (see phase diagram Fig 4C; red-orange colour for most of the initial conditions). For comparison, simulated trajectories and the phase diagram for *ktn1-5* papillae are shown in S3A and S3B Fig and confirm that shape differences between WT and *ktn1-5* papillae have only a minor role in pollen tube growth behaviour. When comparing the simulated and experimental cumulative distributions of turn numbers, a pollen tube growth model incorporating growth alignment with the longitudinal papilla axis with a strength of 0.1 successfully reproduces the experimental turn number distributions observed on WT papillae (orange squares, Fig 4D). For completeness, cumulative distributions for various alignment strengths $\mu$ on WT papillae are presented in S3C Fig and confirms that an alignment strength greater than 0.025 is consistent with the distribution of turn number experimentally observed on WT papillae. In addition, S3D Fig shows the simulated cumulative distribution of turn numbers for *ktn1-5* papillae in the presence of small guidance cues, which confirms that only a very small guidance cue ($0 \leq \mu < 0.01$) reproduces the experimental distributions of pollen tube turn on the mutant papillae.

In conclusion, we demonstrated that a guidance cue within the cylindrical region of the papilla, aligning the pollen tube growth direction along the longitudinal axis of the papilla with a sufficient strength ($\mu > 0.025$), efficiently directs pollen tube growth towards the papilla base and accurately reproduces the straight tube trajectories experimentally observed on WT papillae.

Another significant outcome of our simulations is that the shape variations observed between WT and *ktn1-5* papillae (including neck and head variations, Fig 1H–1K) are not sufficient to globally impact the pollen tube trajectories, regardless of the absence (Fig 3 and S2 Fig) or presence (Fig 4 and S3 Fig) of growth guidance from the papilla side. This conclusion aligns with our previous findings in [6].

## Estimation of the growth guidance through the papilla cell wall

We subsequently explored which mechanism could provide the guidance cue in the longitudinal direction. In a very simplified picture, the papilla can be seen as a quasi-cylinder and the pollen tube as a filament progressing within the elastic shell of this cylinder. Following this picture the deformation of the cylinder surface depends on the tube direction, longitudinal vs circumferential (Fig 5A). Indeed, when the tube grows in a longitudinal direction,

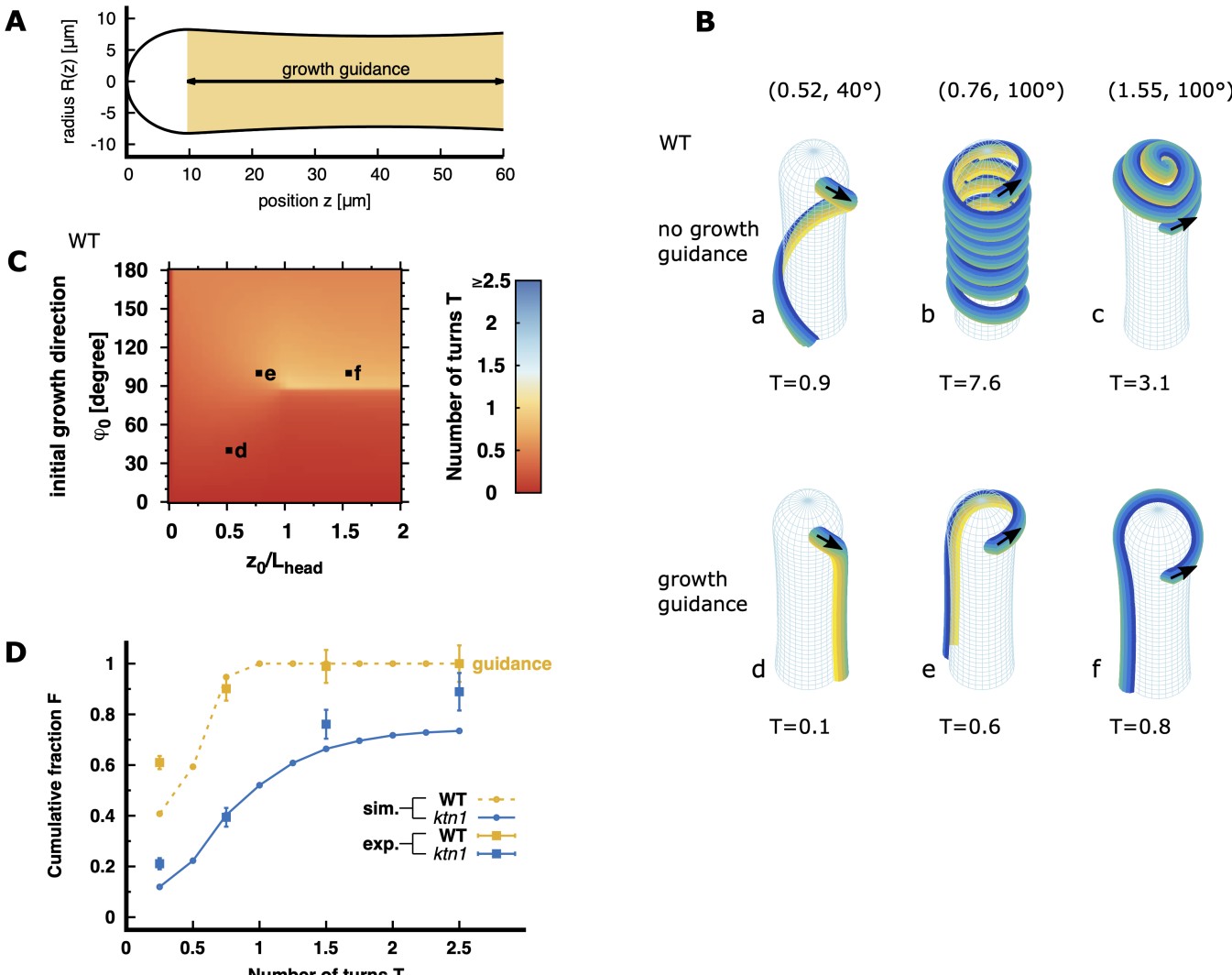

**Fig 4. Effect of longitudinal growth guidance on pollen tube trajectories. (A)** Two-dimensional representation of the papilla, indicating where the alignment forces act (yellow region). **(B)** Example of pollen tube trajectories on WT papilla surfaces under various initial conditions, the initial pollen grain position $z_0$ and initial pollen tube direction $\varphi_0$ (indicated by black arrows), simulated without growth guidance (upper panel) and with growth guidance (adimensional guidance strength $\mu = 0.1$, lower panel). The numbers in brackets denote the normalized initial position ($z_0/L_{head}$) and the initial directions ($\varphi_0$) for each trajectory. T represents the number of turns the pollen tube makes to reach the papilla base. Each configuration is labelled from a to f. **(C)** Morphological phase diagram of the pollen tube turn number $T$ depending on the initial pollen grain position $z_0$ (normalised to the papilla head length $L_{head}$) and the initial pollen tube direction ($\varphi_0$) on WT papillae. $z_0/L_{head} = 0$ denotes the papilla pole, $z_0/L_{head} = 1$ the frontier between the head and cylindrical shaft, and $z_0/L_{head} = 2$ the pollen grain landing limit. The colour code indicates the number of turns the trajectories undergoes before it reaches the papilla base. The letters d, e, f correspond to the example configurations depicted in (B). Trajectories of pollen tubes on *ktn1-5* papilla with identical initial conditions and the corresponding morphological phase diagram are shown in S3 Fig. **(D)** Comparison of simulated (solid lines) and experimental (squares) cumulative distributions of pollen tube turn numbers. Simulated cumulative distributions for turn numbers on WT papillae are calculated with a growth guidance of $\mu = 0.1$ (dashed orange curve). Simulated cumulative distributions for turn numbers on *ktn1-5* papillae are calculated without guidance (blue curve, reproduced from Fig 3E). To calculate the cumulative fraction for experimental data, we utilized data from Ref [6], where we examined 251 WT and 327 *ktn1-5* pollinated papillae; error bars represent the standard error of the mean. The label *ktn1* refers to the *ktn1-5* mutant.

the cylinder surface is solely strained in the circumferential direction, with no strain along the longitudinal axis (Fig 5A, insert b vs. a) while a tube growing in the circumferential direction induces a surface deformation in both directions (Fig 5A, insert c vs. a). Tube growth therefore entails different energetic costs, with longitudinal growth energetically less costly

than circumferential one. We can therefore reasonably assume that this energetic difference might be at the origin of a force on the advancing tube, aligning its growth with the less costly longitudinal direction.

Hence, why does a pollen tube growing on the *ktn1-5* papilla, which bears a similar quasi-cylindrical geometry to WT papillae, not align with the longitudinal papilla direction? Interestingly, in our previous work [6] we found that the mechanical properties of WT and *ktn1-5* papilla cell walls diverged and proposed that these differences could play a key role in pollen tube trajectory. Therefore, here, we investigated whether the mechanical properties of the papilla cell wall, in conjunction with the cylindrical papilla shape, could serve as an effective guidance mechanism for directing pollen tube growth. We considered two main hypotheses. In the first hypothesis, the papilla cell wall is supposed to be mechanically isotropic, but with different rigidities for inner and outer leaflets (*Rigidity contrast hypothesis*). In the second hypothesis we assumed that both wall leaflets have identical rigidity, but are mechanically anisotropic, i.e. rigidity differs depending to the direction within the papilla wall (*Anisotropy hypothesis*).

***Rigidity contrast hypothesis.*** In the first set of simulations, we assumed that the papilla cell wall is isotropic, exhibiting uniform rigidity in all directions within the wall. Thereby, we considered that the inner and outer cell wall leaflets are two-dimensional isotropic elastic sheets characterised by the Young's moduli $Y_{out}$ and $Y_{in}$, respectively. In Riglet et al., we noticed that the passage of the pollen tube deforms the two leaflets of WT and *ktn1-5* papillae cell wall differently [6]. We quantified this deformation (Fig 6B–6D in [6]) and found that in WT papillae, the pollen tube growth results in an almost equal deformation of the cell wall leaflets towards the interior and exterior of the papilla, whereas in *ktn1-5*, the pollen tube deforms the outer cell wall layer to a greater extent than the inner one. Here, we used this deformation quantification to calculate an indentation ratio $\alpha$ corresponding to the ratio between the external ($r_{out}$) and internal ($r_{in}$) papilla deformation. Pollen tube growth on WT and *ktn1-5* papilla results in an indentation ratio $\alpha = 1$ (Fig 5B) and $\alpha = 3$ (Fig 5C), respectively. We leveraged this experimentally determined ratio $\alpha$ to approximate the outer cell wall stiffness ($Y_{out}$) relative to the inner cell wall stiffness ($Y_{in}$) (for details see S1 Text). In the case of WT parameters ($\alpha = 1$), the Young's modulus of the outer cell wall layer is consistently higher than the Young's modulus of the inner cell wall layer (positive rigidity contrast $Y_{out} - Y_{in} > 0$; Fig 5D). Conversely, for *ktn1-5* parameters ($\alpha = 3$), the outer cell wall layer exhibits smaller rigidity than the inner one (negative rigidity contrast $Y_{out} - Y_{in} < 0$; Fig 5D). This may explain why pollen tubes deform the soft outer layer making ridges well visible on the *ktn1-5* papilla surface (Fig 1F and 1G and S4 Fig).

Next, we utilised the estimated Young's moduli $Y_{out}$ and $Y_{in}$ to estimate the strain energy density ($f$) resulting from the stretching of the cell wall due to the tube's passage. Given that the orientation of pollen tube growth (longitudinal vs circumferential) impacts the strain encountered by the papilla cell wall (Fig 5A and as outlined above), we specifically estimated the increase in strain energy induced by the pollen tube growth in the longitudinal ($f_{lg}$) and circumferential ($f_{ci}$) direction (for details see S1 Text).

Finally, the alignment factor $\mu$ acting on the pollen tube tip depends on the difference in strain energies between circumferential and longitudinal tube growth. Indeed, $\mu$ can be estimated as follows: $\mu \approx (f_{ci} - f_{lg})\ell^2/\chi$ where $\ell \sim r$ signifies a typical length scale (such as the pollen tube radius), and $\chi$ represents the bending rigidity of the pollen tube (for more details, see S1 Text). A positive value of $\mu$ indicates an orientation of the pollen tube growth to align with the longitudinal papilla axis, whereas a negative value of $\mu$ favours circumferential tube growth. In our simulations, we found that the alignment factor $\mu$ for WT and *ktn1-5*

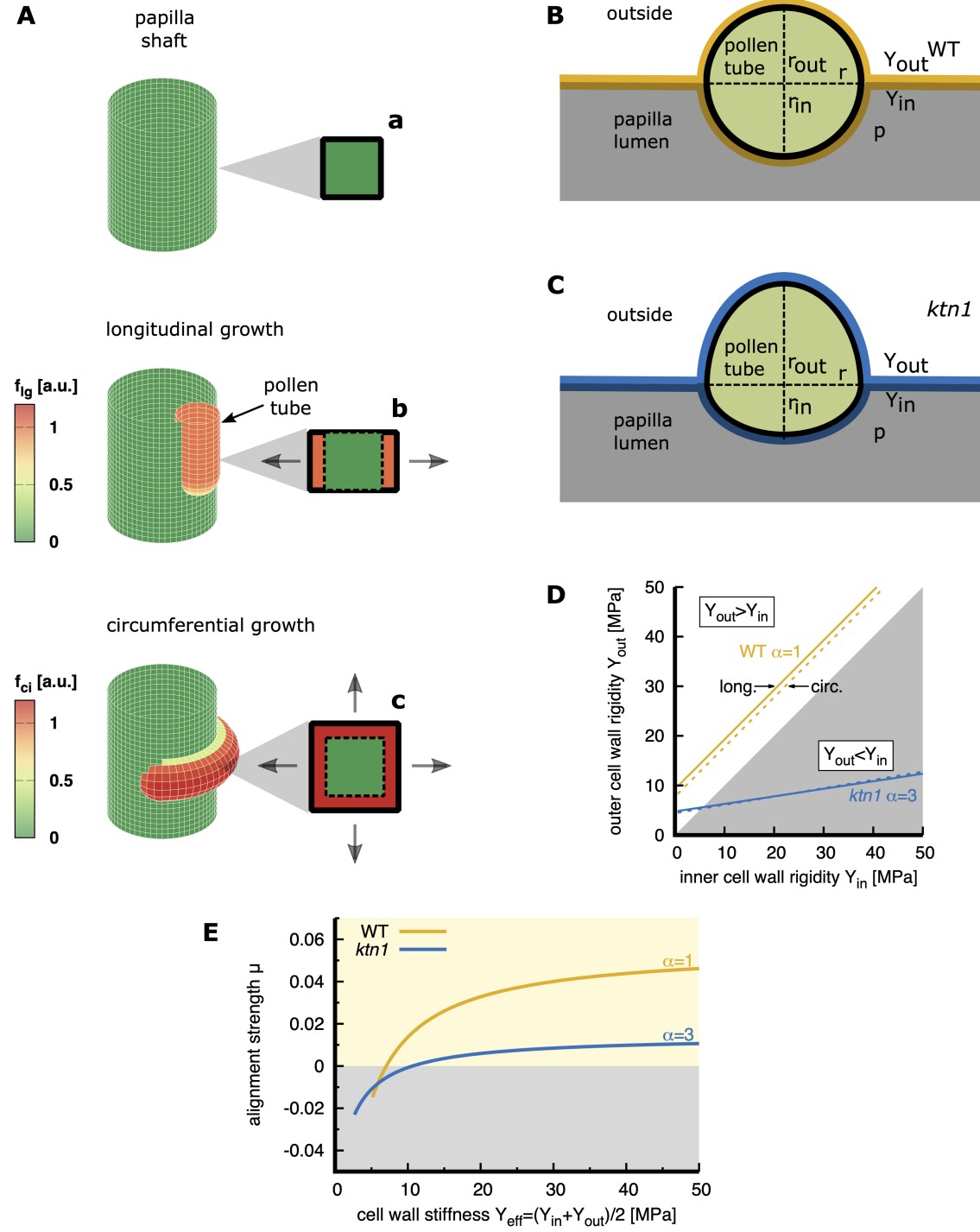

**Fig 5. Model for pollen tube growth guidance assuming a rigidity contrast between the two-dimensional isotropic elastic leaflets of the papilla wall. (A)** The orientation of pollen tube growth (longitudinal *vs.* circumferential) influences the strain experienced by the papilla cell wall. To facilitate the visualisation of the deformation generated by the pollen tube growth, the papilla surface is represented using a grid of circumferential and longitudinal lines. In contrast to the papilla without tube growth, where the grid is made of squares (square, insert a), longitudinal growth generates a deformation causing expansion of the grid in one direction (grey arrows, insert b, and dashed square for non-deformed state). Circumferential growth results in deformation causing the grid to expand in two directions (grey arrows, insert c, dashed square for non-deformed state) The colours represent schematically the local strain energy in the papilla cell wall, $f_{lg}$ and $f_{ci}$, for longitudinal and circumferential growth, respectively. **(B,C)** Mechanical model of pollen tube growth within the WT (B) or the *ktn1-5* (C) papilla cell wall. The pollen tube separates and deforms the cell wall bilayer with an outer

(inner) Young's modulus $Y_{out}$ ($Y_{in}$) and exerts volume work against the papilla pressure $p$. The shape of the deformation cross-section is approximated by two half-ellipses with the indentation ratio $\alpha$ corresponding to the ratio between the external ($r_{out}$) and internal ($r_{in}$) papilla deformation. $\alpha = 1$ for pollen tube growing within the WT papilla cell wall and $\alpha = 3$ for pollen tube growing within *ktn1-5* papilla cell wall. **(D)** Relation between inner and outer cell wall rigidities for a given value of $\alpha = 1$ (WT) and $\alpha = 3$ (*ktn1-5*) for growth in the circumferential (circ.) direction compared to the longitudinal (long.) direction. In the shaded region, the rigidity of the outer cell wall layer $Y_{out}$ is lower than the rigidity of the inner cell wall layer $Y_{in}$ which corresponds to a negative rigidity contrast $Y_{out} < Y_{in}$, otherwise the rigidity contrast is positive $Y_{out} > Y_{in}$. Note that both tube growth directions require a similar rigidity contrast and differences in the indentation ratio $\alpha$ are experimentally probably not detectable. For further details see S1 Text. **(E)** Adimensional alignment strength $\mu$ for WT ($\alpha \approx 1$) and *ktn1-5* ($\alpha \approx 3$) papilla cells depending on the effective cell wall stiffness $Y_{eff} = (Y_{in} + Y_{out})/2$. The label *ktn1* in (D) refers to the *ktn1-5* mutant.

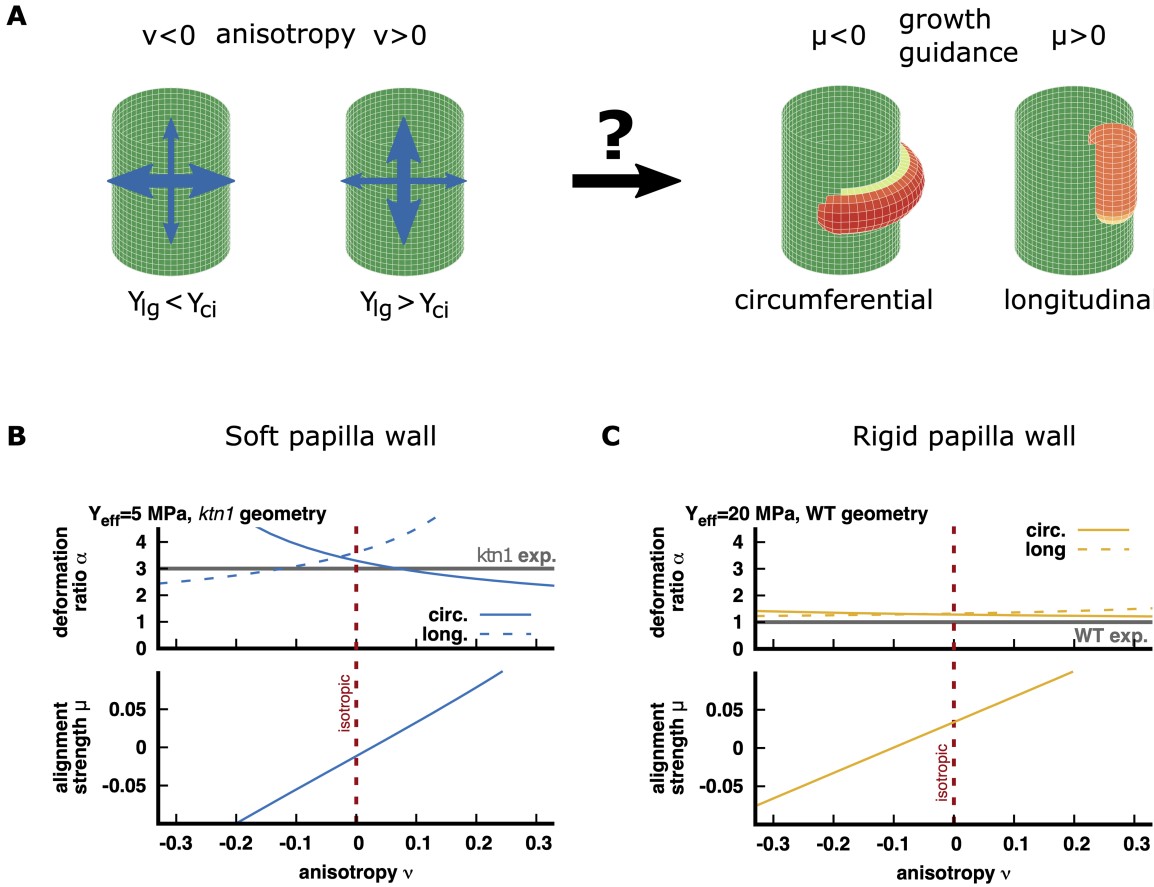

**Fig 6. Role of cell wall mechanical anisotropy in pollen tube guidance. (A)** Schematic representation of the papilla cell wall anisotropy $\nu$ and alignment strength factor $\mu$. $Y_{lg}$ ($Y_{ci}$) stands for the effective stiffness in the longitudinal (circumferential) direction within the papilla wall. Note that we considered $Y_{ci}$ ($Y_{lg}$) to be the same in the inner and outer papilla wall leaflets. When $Y_{lg} < Y_{ci}$, $\nu$ is negative, the longitudinal direction is softer (narrow blue arrow) compared to the circumferential one (large blue arrow). When $Y_{lg} > Y_{ci}$, $\nu$ is positive, the circumferential direction is softer compared to the longitudinal one. A positive value of the alignment strength factor $\mu$ favours a circumferential pollen tube growth whereas a negative value of $\mu$ favours a longitudinal growth. We explored how the anisotropy $\nu$ can influence the alignment strength factor $\mu$ and hence the pollen tube growth direction (question mark). **(B,C)** Indentation ratio $\alpha$ (upper panels) and alignment strength factor $\mu$ (lower panels) for circumferential (circ, solid line) and longitudinal (long, dashed line) tube growth depending on the papilla wall mechanical anisotropy $\nu$ and the effective rigidity of the papilla wall ($Y_{eff} = 5\,MPa$, B: $Y_{eff} = 20\,MPa$, C). The calculations were done using the ratio $\rho$ for *ktn1-5* geometry ($\rho = 2.4\,\mu$m/10 $\mu$m, depicted in blue, B) or for WT geometry ($\rho = 2.4\,\mu$m/7 $\mu$m, depicted in orange, C). The grey lines correspond to the experimentally measured indentation ratios $\alpha = 3$ on *ktn1-5* and $\alpha = 1$ on WT papillae. Additional calculations for *ktn1* geometry with rigid cell wall, WT geometry with soft cell wall and intermediate wall rigidity ($Y_{eff} = 10\,MPa$) are provided in S1 Text. The dashed red line highlights the isotropic case ($\nu = 0$).

papillae depends on the overall cell wall rigidity ($Y_{in} + Y_{out}$)/2 (Fig 5E). Interestingly, in our previous study we experimentally measured the elastic modulus of the papillae cell wall for WT ($Y_{eff} \simeq 18$ MPa) and *ktn1-5* ($Y_{eff} \simeq 25$ MPa) (see Fig 6F in [6]). Within this range of cell wall stiffness, this corresponds to a positive value of the $\mu$ (Fig 5E) for both WT and *ktn1-5* papillae, indicating that the alignment of the tube growth direction with the long papilla axis is favoured in both cases. Our model also showed that the magnitude of the reorientation strength factor $\mu$ depends on the indentation ratio $\alpha$ (Fig 5E). Notably, the factor $\mu$ for *ktn1-5* papillae with an indentation $\alpha = 3$ is lower ($\mu \approx 0.01$) compared to WT papillae ($\alpha = 1$), which reaches approximately $\mu \approx 0.04$. Note that in our simulations depicting pollen tube paths with growth guidance (see previous section "Pollen tube growth on WT papillae requires guidance cues from the papilla side"; S3C Fig, a $\mu$ factor within the same range ($\mu > 0.025$) reproduces the number of pollen tube turns experimentally observed on WT papilla.

Thus, our simulations showed that a alignment factor with a strength comparable to that introduced in our model to accurately reproduce the experimentally observed straight tube trajectories on WT papillae, can emerge within an isotropic papilla cell wall, provided that both cell wall leaflets are sufficiently rigid and exhibit a positive rigidity contrast (i.e. the outer leaflet is more rigid than the inner leaflet).

***Anisotropy hypothesis.*** In a second set of simulations, we introduced an anisotropy $\nu$ in the stiffness of the papilla cell wall between the longitudinal ($Y_{lg}$) and circumferential ($Y_{ci}$) direction (see S1 Text for details)

$$\nu = (Y_{lg} - Y_{ci})/(Y_{lg} + Y_{ci}) \tag{1}$$

A positive value of $\nu$ indicates that the papilla cell was is stiffer in the longitudinal direction than in the circumferential direction, whereas a negative value of $\nu$ indicates that the wall is softer in the longitudinal direction compared to the circumferential one (see schematic representation of $\nu$ in Fig 6A). For example, a value of $\nu = 0.33$ ($\nu = -0.33$) corresponds to a longitudinal stiffness which is double (half) as strong as the circumferential stiffness. In comparison, a value of $\nu = 0.1$ ($\nu = -0.1$) indicates a very weak anisotropy where the longitudinal direction is only about 20 % stiffer (softer) than the circumferential direction. Therefore, we considered that the range of values $-0.1 \leq \nu \leq 0.1$ to define a very weakly anisotropic wall material.

Next, we postulated that the anisotropy $\nu$ and the effective stiffness ($Y_{eff} = (Y_{lg} + Y_{ci})/2$) are identical in the inner and outer leaflets of the papilla cell wall. We then, explored how the indentation ratio $\alpha$ and alignment strength factor $\mu$ vary with the anisotropy $\nu$ for soft ($Y_{eff} = 5$ MPa, Fig 6B) and rigid papilla walls ($Y_{eff} = 20$ MPa, Fig 6C). See also Fig B in the S1 Text for an intermediate papilla wall rigidity ($Y_{eff} = 10$ MPa). To account for the geometrical differences between WT and *ktn1-5* papillae, we performed calculations using the experimentally defined ratio $\rho$ of the pollen tube radius to the papilla radius i.e. $\rho = 2.4 \, \mu$m/7 $\mu$m for WT geometry and $\rho = 2.4 \, \mu$m/10 $\mu$m for *ktn1-5* geometry.

For pollen tubes growing in a soft cell wall, our simulation showed that the indentation ratio $\alpha$ varies with cell wall anisotropy (Fig 6B, upper panel). Note that $\alpha$ also depends on the direction of growth of the tube, the two extreme cases (circular *vs* longitudinal pollen tube growth) being represented in Fig 6B (dashed and solid lines, upper panel). The curves shown were obtained using a *ktn1-5* geometry (blue colour); the calculation using a WT geometry gives very similar results and are shown in Fig B in the S1 Text. In every case, the indentation ratio is well above 1 meaning a greater papilla deformation towards the exterior of the cell as a soft papilla wall ($Y_{eff} = 5$ MPa) poorly resists to the papilla turgor pressure pushing the pollen tube out. The experimentally measured indentation ratio for pollen tube growing in *ktn1-5*

cell wall ($\alpha$ = 3) is reproduced for an isotropic or very weakly anisotropic wall, ($-0.1 \leq \nu \leq 0.1$) (Fig 6B, upper panel). We also observed that for this low rigidity value of the papilla wall ($Y_{eff}$ = 5 MPa) the experimentally measured indentation ratio for pollen tubes growing in WT cell walls ($\alpha$ = 1) can never be observed.

For rigid cell walls, our simulation showed that the indentation ratio $\alpha$ varies only slightly with cell wall anisotropy and does not depend on the main tube growth direction (circular *vs* longitudinal, Fig 6C, upper panel). The curves shown (orange colour) were obtained using the WT geometry; calculation with the *ktn5-1* geometry yielded qualitatively similar results (Fig B in the S1 Text). The calculated $\alpha$ value, which is slightly above 1, reproduced the papilla deformation experimentally measured for pollen tubes growing in WT cell walls ($\alpha$ = 1), but failed to replicate the deformation measured in *ktn1-5* cell walls ($\alpha$ = 3).

Overall, the indentation ratio for pollen tube growth observed in the *ktn1-5* wall ($\alpha$ = 3) can only be reproduced in our model assuming isotropic, or very weakly anisotropic, soft cell wall. It can be noticed that to reach the value $\alpha$ = 3 a significant decrease of the effective stiffness of the papilla cell wall ($Y_{eff} \leq 5$ MPa) is required (see also Fig B in the S1 Text). By contrast, the indentation ratio experimentally measured for pollen tube growth in the WT wall ($\alpha$ = 1) can only be reproduced in our model assuming a rigid cell wall, independently of the wall anisotropy.

For both soft and rigid cell walls, our simulations showed that the alignment strength factor $\mu$ varies quasi-linearly with the cell wall anisotropy (Fig 6B and 1C, lower panels). For soft cell walls (Fig 6B, lower panel), a positive $\mu$ value, indicating that pollen tube growth aligns with the longitudinal papilla axis, is only achieved for a positive value of the anisotropy $\nu$ (stronger rigidity in the wall longitudinal direction, see schematic representation of $\mu$ and $\nu$ in Fig 6A). For a rigid cell wall (Fig 6C, lower panel), a positive $\mu$ value can also be achieved for weakly negative anisotropy values of $\nu$ ($-0.1 \leq \nu \leq 0$), indicating that a rigid cell wall with slightly greater stiffness in the circumferential direction than in the longitudinal direction can support longitudinal pollen tube growth. However, it is worth noting that for these small negative $\nu$ values ($-0.1 \leq \nu \leq 0$), the guiding factor $\mu$ remains below 0.025. Overall, our simulation showed that to reach a sufficient alignment strength, consistent with the straight tube trajectories experimentally observed on WT papillae ($\mu$ > 0.025; S3C Fig, requires a positive cell wall anisotropy ($\nu \geq 0$), regardless of the cell wall's rigidity.

## Discussion

The plant epidermis fulfils a broad range of functions, with cells of diverse shapes and sizes specialized for their specific roles [18]. Among its functions, the stigmatic epidermis, featuring hundreds of quasi-cylindrical papillae, serves as a receptive platform for pollen grains, facilitating pollen capture, a critical first step in ensuring successful reproduction [19].

Riglet and colleagues [6] have recently underscored the significant role of mechanical properties of the stigmatic cell wall in influencing pollen tube growth directionality. Yet, distinguishing the relative contribution of papilla parameters, such as its wall mechanical properties or its cylindrical geometry, on tube guidance remains a significant technical challenge, especially due to the intricate interplay between cell shape development and cell wall characteristics. We thus decided to take a theoretical approach and to assess how our mechanical modelling aligns with experimental data.

We developed a mechanical model to simulate pollen tube trajectories on the papilla surface. Our simulations showed that when we introduce a sufficiently strong guidance cue ($\mu$ > 0.025, Fig 4D and S3C Fig) within the cylindrical region of the papilla, simulated trajectories align along the papilla longitudinal axis, reproducing the tube behaviour observed

experimentally on WT papillae. In contrast, when the guidance cue is weak or absent from the papilla side ($0 \leq \mu < 0.01$, Fig 3E and S3D Fig), simulated trajectories follow curved close to geodesic, mimicking the coiling behaviour experimentally observed on *ktn1-5* papillae.

We then hypothesised that two key features of the interaction between the papilla and the pollen tube tip, i.e. the quasi-cylindrical shape of papilla and the stretching of the papilla wall due to the tube passage, could together generate the effective guidance cue. In this context, we tested two main hypotheses.

*1. Rigidity contrast hypothesis*: We used the experimentally measured deformation of the papilla cell wall (indentation factor $\alpha$) to estimate the contrast in rigidity between the inner and outer isotropic leaflets of the papilla wall. We, then, performed an order of magnitude calculation of the energetic cost of stretching both leaflets, depending on the orientation of tube growth (longitudinal vs circumferential, Fig 5A). From this energetic difference, we deduced the magnitude of the alignment strength $\mu$ acting on the pollen tube tip. For an indentation factor $\alpha$ of 1, our calculation showed that the outer cell wall layer is consistently more rigid that the inner one (Fig 5D) which, in the stiffness range experimentally measured for WT cell wall ([6]), results in an alignment strength $\mu > 0.025$ (Fig 5E), compatible with longitudinal tube growth. Conversely, for an indentation factor $\alpha$ of 3, our calculations indicate that the outer layer is softer than the inner one (Fig 5D) resulting in a lower alignment strength ($\mu < 0.01$; Fig 5E) compatible with the coiling behaviour of pollen tube growth. Thus, our simulation showed that an isotropic cell wall with a sufficiently rigid outer layer to resist against the papilla turgor pressure could provide a guidance cue to align the pollen tube growth with the longitudinal direction of the papilla as observed in the WT cell wall.

*2. Anisotropy hypothesis:* The plant cell wall is a complex visco-elastic material, composed of cross-linked cellulose microfibrils embedded in a polymeric matrix, which may exhibit anisotropic mechanical properties [20,21]. To account for this, we incorporated a direction-dependent mechanical rigidity in the papilla wall in our model. Our simulations showed, first, that to reproduce the parameters leading to coiled trajectories in *ktn1-5* cell wall (i.e. indentation ratio $\alpha = 3$ and a alignment strength $0 \leq \mu < 0.01$), an isotropic ($\nu \approx 0$) and soft ($Y_{eff} < 5$ MPa) cell wall is required (Fig 6B). In addition, we found that for higher rigidities ($Y_{eff} = 10$ MPa or 20 MPa), within the range experimentally measured for *ktn1-5* cell wall ([6]), an indentation ratio $\alpha$ of 3 can never be observed. Therefore, our model suggests that incorporating anisotropy in the papilla wall is not compatible with the coiling behaviour of the pollen tube growing in a *ktn1-5* cell wall. Second, we found that reproducing the parameters leading to the pollen tube behaviour in WT cell wall (indentation ratio $\alpha = 1$ and positive alignment strength $\mu$) would require a rigid cell wall ($\geq 20$ MPa; Fig 6C) consistent with our previous measurements of WT cell wall rigidity ([6]). In addition, we found that a sufficient alignment strength ($\mu > 0.025$) compatible with the longitudinal tube growth, can be obtained for an isotropic cell wall ($\nu = 0$), consistent with our results for isotropic cell walls with a rigidity contrast between the outer and inner leaflets (Fig 5E), or a cell wall with positive anisotropic ($\nu > 0$; Fig 6C). This latter observation means that the pollen tube would progress in a wall with longitudinal direction stiffer than the circumferential one. This finding contrasts with the conventional knowledge regarding walled pressurized cylindrical structures (as a papilla) where the wall strengthens the structure in the circumferential direction leading to increased circumferential rigidity compared to the longitudinal direction [22–25]. Hence, a direction-dependent mechanical rigidity in the papilla wall appears incompatible with the longitudinal orientation of pollen tube growth in the WT cell wall. Quantitative information on cell wall anisotropy, along with more advanced calculations, that incorporate the complex cell wall rheology and pollen tube growth dynamics, are now required to fully elucidate

the role of cell wall anisotropy in pollen tube growth guidance and will be the focus of future work.

At the current state of knowledge, we therefore consider it is more likely, that a rigidity contrast between the inner and outer leaflets contributes to establish different indentation ratios on WT and *ktn1-5* papillae. Altogether, this work pinpoints the critical role of the papilla cylindrical geometry and the rigidity contrast between the two wall layers in providing a guidance cue. This cue, if sufficiently strong, can deviate the tube tip from geodesics and favours growth along the less costly longitudinal direction of the cylindrical papilla, thereby orienting the pollen tube towards its base. In contrast, when the tube is facing less constraints to deform the outer wall layer in *ktn1-5* cell wall, the growth guidance exerted on the advancing tube tip may not be sufficient to deviate the pollen tube away from geodesic paths, thereby leading to the observed coiling behaviour (Fig 7).

In our prior research [6], we examined the turning behaviour of pollen tubes as they grew on papillae of cell wall mutants, notably focusing on the two mutants *xxt1xxt2* (affected in

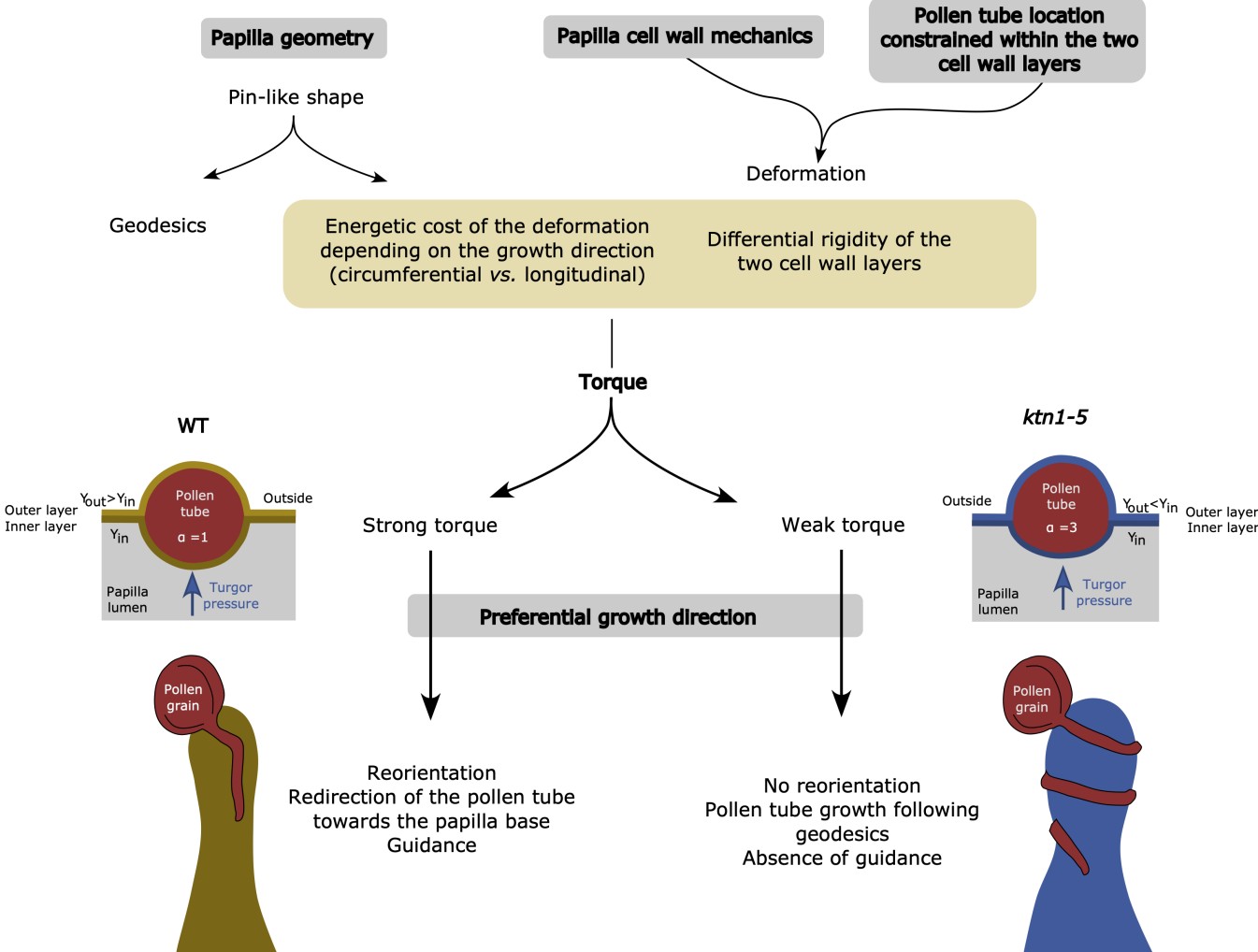

**Fig 7. Papilla geometry and cell wall mechanics act synergistically to orient the advancing pollen tube along the longitudinal papilla axis.**

hemicellulose synthesis) and *any1* (impaired in the cellulose synthase complex). We found that pollen tubes predominantly grow in a straight manner on these mutant papillae, similar to their growth on WT. Visual inspection of several SEMi images revealed that pollen tubes growing on *xxt1xxt2* and *any1* mutant papillae, while showing varying degrees of bulging, were not as prominent as those observed on *ktn1-5* papillae (S4 Fig). Hence, it is reasonable to expect that the indentation ratio $\alpha$ for both mutant papillae would more closely resemble that of WT papillae. Consequently, we can presume that the inferred alignment factor ($\mu$) would be of a similar magnitude to that calculated for WT papillae, guiding the pollen tube to align with the papilla's longitudinal axis. Estimation of the $\mu$ factor for *xxt1xxt2* and *any1* requires a meticulous quantification of papilla deformation, aligning with the goals of our forthcoming research. Nonetheless, our analysis of SEMi images suggests the broader applicability of our model to mutants beyond *ktn1-5*.

Our findings highlight a plausible function of the remarkable ability of the pollen tubes to navigate engulfed within the two layers of the stigmatic cell wall, a capability that stands out especially when compared with other invasive organisms such as filamentous pathogens [26]. It was previously hypothesized that the bilayer cell wall structure, not observed in other cell types of the plant body, may represent a specialized adaptation to accommodate pollen-tube growth [5]. Our work strengthens this hypothesis as we showed that maintaining the pollen tube at the papilla surface between two elastic layers with different rigidities represents an efficient way to control the directionality of the growing pollen tube by making use of the papilla geometry.

Interestingly, the semi-aquatic genus Trithuria, an early-divergent angiosperm, has a dry stigma like Arabidopsis, composed of long multicellular stigmatic hairs and the pollen tubes grow within the two distinct cell wall layers of these cells, mainly straight toward the base [27]. Dry stigmas have been suggested to represent the ancestral condition in Angiosperms [28], which raises the questions whether growing confined within the cell wall, a specificity of the pollen tube behaviour, may also represent the ancestral state for angiosperms.

Chemical gradients are known to be involved in late-stage guidance of pollen tube to ovules [29]. Interestingly, our findings indicate that a polar chemical gradient is not necessary for directing the pollen tube towards the base of the papilla cell. Instead, our work demonstrates that the geometry and mechanical properties of the papilla alone can produce sufficient cues for this. The specific geometry of the papilla cells is tightly linked to their functions of capturing pollen grains. Indeed, successful pollination, i.e. pollen capture, later results in papilla shrinkage, while papillae continue to elongate in the absence of pollination [30]. This suggests that the shape of the papilla represents an effective balance between the need for elongated cells to capture pollen grains over a wide surface area and a quasi-cylindrical geometry to efficiently guide pollen tubes straight down along the papillae.

## Materials and methods

### Biological material and culture conditions

All *Arabidopsis thaliana* lines were in the Col-0 background and grown in growth chambers under long-day conditions (16 h light/8 h dark at 21°C/19°C with a relative humidity around 60 %). *ktn1-5* (SAIL_343_D12) was described previously [6]. All stigmas were analyzed at stage 13 of flower development [31].

## Measurement of shape parameters of papilla cells

Stigma were observed with an upright optical microscope Zeiss Axioimager Z1, under bright field, using a Plan-Apochromat 10x objective. Images processing and dimension measurements (Fig 1J and 1K and S2 Table) were performed using the segmented line and the measure tools of the ImageJ/Fiji software [32].

## Geometric model of a pin-like surface

In Fig 2, we represented pin-like shapes as surfaces of revolution. These surfaces were formalized by defining a profile curve represented as a NURBS curve $N(u^1)$ [33], $u^1$ being a free real parameter between 0 and 1, that is then rotated around a vertical axis according to the equation

$$\mathbf{S}(u^1, u^2) = \begin{pmatrix} N(u^1)\cos(u^2)W \\ N(u^1)\sin(u^2)W \\ u^1 H \end{pmatrix}. \tag{2}$$

$\mathbf{S}(u^1, u^2)$ represents a point at the surface, corresponding to parameters $(u^1, u^2)$. Here, $u^1$ represents a normalised $z$ coordinate - between 0 and 1 - pointing upward and $u^2$ represents the azimuthal coordinate varying in $[0, 2\pi)$. The NURBS curve $N(u^1)$ is defined using a set of control points and scales in width and height, respectively, denoted by scalars $W$ and $H$. Different surface profiles with varying radii at mid-height (referred to as 'the neck') of the model were obtained by changing the most central control point CP3 (see S3 Table).

## Geodesics

Geodesic curves on $\mathbf{S}$ are defined by a set of two second order, non-linear, coupled differential equations with $\alpha = \{1, 2\}$ (see Refs [9] and S1 Text):

$$\frac{d^2 u^\alpha}{ds^2} + \sum_\beta \sum_\gamma \Gamma_{\beta\gamma}^\alpha \frac{du^\beta}{ds} \frac{du^\gamma}{ds} = 0, \tag{3}$$

where $\Gamma_{\beta\gamma}^\alpha$ are the Christoffel symbols of the second kind computed from the first and second derivatives of the surface equation (2) and from scalar products between these. Using Eqs (3), it is possible to compute geodesic trajectories on the surface from given initial conditions corresponding to some initial position $P_0 = [u^1, u^2]_{s=0}^T$ and orientation $\mathbf{t}_0 = \left[\frac{\partial u^1}{\partial s}, \frac{\partial u^2}{\partial s}\right]_{s=0}^T$ on the surface. In Fig 2 we used $P_0 = [0, -0.1]^T$ (the point is slightly below the tip) and an initial inclination is $\mathbf{t}_0 = [6.7, -0.47]^T$ corresponding to an angle of 28° downwards with respect to the horizontal. To integrate these equations we used the odeint function of the Python SciPy library [34]. For the simulations of geodesics in Fig2, we used our new software package to simulate L-systems in Riemannian spaces. The information related to this package is documented in Ref [11]. The numerical code was written using the Riemannian-LPy language, available at [35]. Data were visualised using the same software. Fig 2 can be generated with the code in [36] and the control point values and parameters described in S3 Table.

## Analytical expressions for virtual papilla shapes

To proceed further with the geometric model of the papilla surface and relate its parameters to geometric quantities that can be measured on the images, we designed a variant of the previous parametric expression of the surface, where the dependency of the papilla radius $R$ on the

$z$ coordinate is expressed by an explicit formula with easily interpretable geometric parameters. Note that we introduce here a short notation, i.e. $L_h = L_{\text{head}}$, $L_n = L_{\text{neck}}$, $W_h = W_{\text{head}}$, and $W_n = W_{\text{neck}}$. As in the previous section, we expressed the rotational symmetry of the papilla surface $\mathbf{S}$ using a surface of revolution with cylindrical coordinates $(\theta, z)$ as:

$$\mathbf{S}(\theta, z) = \begin{pmatrix} R(z)\cos(\theta) \\ R(z)\sin(\theta) \\ z \end{pmatrix} \tag{4}$$

with the papilla long axis oriented along $z$. However, the radius function $R(z)$ is now given by:

$$R(z) = \begin{cases} \frac{W_h}{2}\sqrt{1 - \frac{(L_h - z)^2}{L_h^2}} & \text{for} \quad 0 \leq z \leq L_h \\ \frac{W_h}{2} - \frac{W_h}{4L_h^2}(L_h - z)^2 + \frac{a_1(z - L_h)^3}{a_2 + z - L_h} & \text{for} \quad L_h < z \end{cases}, \tag{5}$$

where the parameters $L_h$ and $W_h$ can readily be measured on the images, (cf. Fig 1F and 1G). Note that the papilla head is described as an ellipsoid. At $z = L_h$ the radius function is continuous up to the second derivative of $R$ w.r.t. $z$. The shape parameters $a_1$ and $a_2$ were adapted to fulfill the following conditions (cf. Fig 1F and 1G)

$$\frac{dR}{dz} = 0 \quad \text{at} \quad z = L_n \tag{6}$$

$$R(L_n) = \frac{W_n}{2} \tag{7}$$

and can be calculated from the following analytical expressions

$$a_1 = -\frac{(2L_h L_n W_h + L_h^2 W_h - L_n^2 W_h - 2L_h^2 W_n)^2}{4L_h^2(L_h - L_n)^2(2L_h L_n W_h + 5L_h^2 W_h - L_n^2 W_h - 6L_h^2 W_n)} \tag{8}$$

$$a_2 = \frac{4L_h^2(L_h - L_n)(W_h - W_n)}{2L_h L_n W_h + 5L_h^2 W_h - L_n^2 W_h - 6L_h^2 W_n}. \tag{9}$$

### Numerical details of the phenomenological model for pollen tube trajectories

The pollen tube path $\mathbf{X}(s)$ is discretized into equidistant points $\mathbf{X}(i\delta s)$, where $s$ denotes the arc length. In the cylindrical region of the papilla [i.e. where the surface can be conveniently parametized in cylindrical coordinates $(\theta, z)$, see Eq (4)] the pollen tube path is given by the following ordinary differential equation

$$\frac{d\mathbf{X}}{ds} = \mathbf{t} = \cos(\varphi)\,\hat{\mathbf{t}}_1 + \sin(\varphi)\,\hat{\mathbf{t}}_2. \tag{10}$$

$\varphi$ denotes the angle between the tip tangent $\mathbf{t}$ and the surface tangent $\hat{\mathbf{t}}_1$ pointing along the long papilla axis

$$\hat{\mathbf{t}}_1 = \left(R'\cos\theta\,\hat{\mathbf{e}}_\mathbf{x} + R'\sin\theta\,\hat{\mathbf{e}}_\mathbf{y} + \hat{\mathbf{e}}_\mathbf{z}\right)\frac{1}{\sqrt{R'^2 + 1}} \tag{11}$$

with $R' = dR/dz$ denoting the first derivative of the papilla radius w.r.t. $z$. In the here chosen definition of $\varphi$, an angle $\varphi = 0°$ ($\varphi = 180°$) indicates a pollen tube tip oriented towards the papilla base (pole).

The second surface tangent $\hat{\mathbf{t}}_2$ is oriented in the circumferential direction of the papilla surface and forms an orthonormal basis with $\hat{\mathbf{t}}_1$

$$\hat{\mathbf{t}}_2 = -\sin\theta\,\hat{\mathbf{e}}_\mathbf{x} + \cos\theta\,\hat{\mathbf{e}}_\mathbf{y}. \tag{12}$$

The momentum conservation at the pollen tube tip is given by the following ordinary differential equation and determines the evolution of the angle $\varphi(s)$ along the pollen tube trajectory

$$\chi\left[\frac{d\varphi}{ds} + \frac{\sin\varphi R'}{R\sqrt{1+R'^2}}\right] + \mathcal{M} = 0, \tag{13}$$

where $\chi$ denotes a bending rigidity and where

$$\mathcal{M} = m\sin(2\varphi) \quad \text{for} \quad z > z_c. \tag{14}$$

denotes the external torque of magnitude $m$ which aligns the growth direction (described by $\varphi$) with the long papilla axis for $z > z_c$. Here we chose $z_c = A$, where the almost spherical cap of the papilla goes over to the cylindrical shaft. Note that expression (14) favors growth along the longitudinal direction. Depending on the angle $\varphi$ the preferred growth direction is towards the papilla pole or the papilla base.

Eq (13) can be derived from a variational principle considering the following energy functional for the tube bending energy

$$\mathcal{F} = \int_0^L \frac{\chi}{2}\left(\frac{\partial t}{\partial s}\right)^2 ds - W, \tag{15}$$

where $W$ denotes the work performed by external forces on the tube tip. Variation of $\mathcal{F}$ w.r.t. the tube tangent vector $\mathbf{t}$ results in

$$\delta\mathcal{F} = \int_0^L -\chi\frac{\partial^3\mathbf{t}}{\partial s^3}.\delta\mathbf{t}\,ds + \left(\chi\frac{\partial\mathbf{t}}{\partial s} - \mathbf{T}\right).\delta\mathbf{t}\Big|_0^L \tag{16}$$

The bulk term and the boundary term at $s = 0$ in (16) vanishes since the tube cannot change its position after it has been deposited. With $\mathbf{T} = 2m(\mathbf{t}.\mathbf{t}_1)\mathbf{t}_1$ and using cylindrical coordinates we recover Eqs (13) and (14).

To express the alignment strength $m$ compared to the tube bending rigidity $\chi$ we introduce the adimensional growth guidance $\mu = m\ell/\chi$, with $\ell$ being the typical length scale (here $\ell \sim 2.5\,\mu$m, corresponding to the pollen tube radius). In the absence of any external torque, i.e. $\mathcal{M} = 0$, Eqs (10) and (13) describe a geodesic line on the papilla surface $\mathbf{S}$.

In the head region of the papilla, i.e. $z \ll A$ the parametrization of the tube path in cylindrical coordinates will fail and we have parametrized the papilla surface in Cartesian coordinates $(x,y)$. Further details on this alternative parametrization as well as on the the implementation of the self-avoidance are given in the S1 Text.

Eq (13) (with self-avoidance) was solved numerically on virtual papilla surfaces representing either WT or *ktn1-5* papillae (see previous section) using a custom written C-code [37]. Data were visualized using the gnuplot package (version 5.2 [38]).

## Calculation of cumulative distributions of turn numbers

To calculate the cumulative distributions of turn numbers $T$ from morphological phase diagrams (Figs 3 and 4) the turn number of each initial attachment site $z_0$ is weighted in the cumulative distribution with a factor $dS = 2\pi R(z)\sqrt{1 + R'(z)^2}\,dz$ corresponding to the $z$-dependent surface area element of the papilla. Trajectories which could not reach the papilla base due to self-avoidance were always counted as trajectories with high turn numbers (T>2.5). Typically, morphological phase diagrams where calculated for increments in the starting angle $\Delta\varphi_0 = 5°$ and the starting position $\Delta z_0 = 5\,\mu$m.

To test to what extend the assumption of an equal probability for pollen grains to attach at the papilla surface up to a height z=2A could flaw the distribution of turn numbers, we tested an alternative method to calculate turn numbers from morphological phase diagrams. There, we used the assumption that pollen grains attach to the papilla surface with a probability derived from the projected area of the papilla head (for $z \leq L_{head}$) in the $x$–$y$ plane resulting in a weight $dS = 2\pi R(z)dR$, where $R(z)$ denotes the $z$–dependent papilla radius in cylindrical coordinates. This method favors attachment at the papilla pole and reduces the contribution of papilla grains attached rather to the side of the papilla head in the cumulative histogram. S3E Fig compares the cumulative turn number distributions for WT ($\mu = 0.1$) and *ktn1-5* ($\mu = 0$) papillae (solid lines similar to Fig 4D) which were obtained with a method using the surface area and the projected surface area as weighing function. The curves (projected and area) are slightly different, but the choice of the weighing function has overall only a small effect on the turn number distribution.

## Supporting information

**S1 Text. Supporting Methods and Materials.** This supplementary text contains:

- Mathematical description of geodesic curves on a surface **S**
- Numerical description of pollen tube trajectories in the papilla pole region
- Numerical implementation of self-avoidance
- A simple mechanical model for the estimation of the mechanical growth guidance through the papilla cell wall.

(PDF)

**S1 Fig. Self avoidance in growing pollen tubes. SEMi images of *ktn1-5* stigmas pollinated with WT pollen.** Pollen grains and pollen tube paths were highlighted in colours for a enhanced visualisation. As the pollen tube produced a marked bump on the *ktn1-5* papilla [6], the tube path was easier to follow and thus the self-avoidance property easier to observe. **(A)** Self-avoidance property: the pollen tube cannot cross its own path. **(B)** When multiple pollen grains germinate on a single *ktn1-5* papilla, their tubes cannot intersect each other. Scale bar = 10 $\mu$m.
(TIF)

**S2 Fig. Examples of pollen tube trajectories on virtual WT papilla surfaces.** The initial positions of the pollen grains $z_0$ and the initial directions of the emerging pollen tubes $\varphi_0$ (indicated by black arrows) match the initial conditions used for simulated trajectories on *ktn1-5* papillae shown in Fig 3C in the main text. The numbers in brackets denote the normalized initial position ($z_0/L_{head}$) and the initial direction ($\varphi_0$) for each trajectory. T (below each

papilla) stands for the number of turns made by the pollen tube to reach the papilla base. Each configuration is labelled from a to h.
(TIFF)

**S3 Fig. Effect of growth guidance on pollen tube trajectories on virtual WT and *ktn1-5* surfaces. (A)** Examples of pollen tube trajectories on virtual *ktn1-5* papillae, varying in initial positions $z_0$ of the pollen grains and initial directions $\varphi_0$ of the emerging pollen tubes (indicated by black arrows), simulated without growth guidance (top) and with growth guidance (adimensional guidance strength $\mu = 0.1$, bottom). Initial conditions ($z_0$ and $\varphi_0$) match those used for simulated trajectories on WT papillae shown in Fig 4B in the main text. The numbers in brackets denote the normalized initial position ($z_0/L_{\text{head}}$) and the initial direction ($\varphi_0$) for each trajectory. T (below each papilla) represents the number of turns the pollen tube makes to reach the papilla base. Each configuration is labelled from a to f. **(B)** Morphological phase diagram for the pollen tube turn number T with growth guidance (adimensional guidance strength $\mu = 0.1$) depending on the normalized initial position of the pollen grain ($z_0/L_{\text{head}}$) and the initial pollen tube direction ($\varphi_0$), on *ktn1-5* papillae. $z_0/L_{\text{head}} = 0$ denotes the papilla pole, $z_0/L_{\text{head}} = 1$ the frontier between the head and cylindrical shaft, and $z_0/L_{\text{head}} = 2$ the pollen grain landing limit. The colour code indicates the number of turns T the trajectories undergoes before it reaches the papilla base. The letters d, e, f correspond to the example configurations depicted in (A). **(C)** Comparison of simulated (solid lines) and experimental (squares) cumulative distributions of pollen tube turn numbers. Simulated cumulative distributions for turn numbers on WT papillae are calculated with increasing growth guidance strength $\mu$ (orange curves). **(D)** Comparison of simulated (solid lines) and experimental (squares) cumulative distributions of pollen tube turn numbers. Simulated cumulative distributions for turn numbers on *ktn1-5* papillae are calculated with increasing growth guidance strength $\mu$ (blue curves). **(E)** Comparison of two different weighing functions to calculate cumulative distributions of pollen tube turn numbers (for details on these two calculation methods see Materials and Methods). Experimental and simulated cumulative distributions of pollen tube turn numbers on WT (guidance strength $\mu = 0.1$) and *ktn1-5* papilla (without guidance) obtained using the surface area ("area", solid lines) or the projected surface area ("proj.", dashed lines) of the papilla. For the "area" function, each point on the papilla surface up to the length $z_0 = 2L_{head}$ receives pollen grains with equal probability. For the "proj." function, the probability of a pollen grain to land up to a length $z_0 = L_{head}$ is weighted by the projected surface area in the x-y-plane. The turn numbers corresponding to the initial conditions ($z_0$, $\varphi_0$) were deduced from the morphological phase diagrams in Fig 3D bottom and Fig 4C. To calculate the experimental cumulative fraction in (C,D,E), we utilized data from Ref [6], where we examined 251 WT and 327 ktn1-5 pollinated papillae. The error bars correspond to the standard error of the mean. The label *ktn1* refers to the *ktn1-5* mutant.
(TIFF)

**S4 Fig. Pollen tube behaviour on papillae of cell wall mutants.** SEMi images of WT, *ktn1-5*, *any1* (impaired in the cellulose synthase complex) and *xxt1xxt2* (impaired in hemicellulose biosynthesis) papillae pollinated with WT pollen grains. Two pollinated stigmas representative of the 12 independent *any1* stigmas and 14 independent *xxt1xxt2* stigmas are shown. Scale bar = 10 $\mu$m.
(TIF)

**S1 Table. Main parameters used in this study and their biological/physical interpretation.**
(PDF)

**S2 Table. Measurements of WT and *ktn1-5*papilla dimensions on images obtained by optical microscopy.**
(PDF)

**S3 Table. Control point coordinates for the NURBS curves used to define the profile curves of the surface of revolution representing the pin-like structures of Fig 2 in the main text.**
(PDF)

## Acknowledgments

We thank Bahram Houchmandzadeh for fruitfull discussions on global minimisation problems. We thank Jonathan Prevot for preliminary *in vitro* pollen tube growth experiments and Valentin Poncet for preliminary calculations of a pollen tube growth model. The computations were performed using the Cactus cluster which is part of the GRICAD infrastructure (https://gricad.univ-grenoble-alpes.fr), which is supported by the Grenoble research communities. The authors thank Philippe Beys who manages the cluster. We thank all the members of the Cell Signaling and Endocytosis (SiCE) group (Laboratoire de Reproduction et Developpement des Plantes, ENS Lyon, France) for fruitful discussions. We thank P. Bolland, A Lacroix, J. Berger (RDP) for plant care.

## Author contributions

**Conceptualization:** Catherine Quilliet, Christophe Godin, Karin John, Isabelle Fobis-Loisy.

**Data curation:** Lucie Riglet, Christophe Godin, Karin John, Isabelle Fobis-Loisy.

**Formal analysis:** Lucie Riglet, Catherine Quilliet, Christophe Godin, Karin John, Isabelle Fobis-Loisy.

**Funding acquisition:** Karin John, Isabelle Fobis-Loisy.

**Investigation:** Christophe Godin, Karin John, Isabelle Fobis-Loisy.

**Methodology:** Lucie Riglet, Catherine Quilliet, Christophe Godin, Karin John, Isabelle Fobis-Loisy.

**Project administration:** Karin John, Isabelle Fobis-Loisy.

**Supervision:** Isabelle Fobis-Loisy.

**Validation:** Christophe Godin, Karin John, Isabelle Fobis-Loisy.

**Visualization:** Christophe Godin, Karin John, Isabelle Fobis-Loisy.

**Writing – original draft:** Lucie Riglet, Catherine Quilliet, Christophe Godin, Karin John, Isabelle Fobis-Loisy.

**Writing – review & editing:** Christophe Godin, Karin John, Isabelle Fobis-Loisy.

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
