## [Decision Letter · Decision Letter 0]

1 Aug 2024

Dear Dr Fobis-Loisy,

Thank you very much for submitting your manuscript "Geometric and mechanical guidance: role of stigmatic

epidermis in early pollen tube pathfinding in Arabidopsis" for consideration at PLOS Computational Biology.

As with all papers reviewed by the journal, your manuscript was reviewed by members of the editorial board and by several independent reviewers. In light of the reviews (below this email), we would like to invite the resubmission of a significantly-revised version that takes into account the reviewers' comments.

As you can see, the four reviewers are generally positive about your manuscript, but they also raise a number of concerns. One of the reviewers suggests to lessen the claim that in the ktn1 mutant the pollen tubes follow geodesics, as these are not uniquely defined at concave parts of the growth surface and become highly variable under shape uncertainties. Two of the reviewers discuss the role of anisotropy and the possibility of including other mutants in your analysis. One of the reviewers seems to understand parameter alpha as a measure of anisotropy,  whereas another reviewer suggests that anisotropy of the cell wall could be an equally important factor, besides stiffness differences, for pollen tube guidance. This issue could be addressed with further simulations or with an extensive discussion, also clarifying the relation with your previous work. A minor remark from my side is that I was confused by the somewhat unusual abbreviation 'CW' for cell wall (as CW is also often used to mean 'clockwise'); I would suggest to expand the word throughout the text.  Please address these and the other comments from the reviewers in a revised version of the manuscript. Please also provide the code used for the simulations as per the requirements of PLOS Computational Biology

We cannot make any decision about publication until we have seen the revised manuscript and your response to the reviewers' comments. Your revised manuscript is also likely to be sent to reviewers for further evaluation.

Sincerely,

Roeland M.H. Merks, Ph.D

Academic Editor

PLOS Computational Biology

Jason Haugh

Section Editor

PLOS Computational Biology

Reviewer's Responses to Questions

**Comments to the Authors:**

Reviewer #1: The present manuscript provides a computational model to simulate the growth of pollen tube on the surface of stigmatic papilla to assess how papilla geometry and the growth pattern between cell wall layers influences the direction of pollen tube growth on its way towards the ovule. The model builds on experimental findings described in a previous paper as well as new findings.

In short - this is a beautiful paper. The model is simple and elegant. The predictions are compelling and convincing. The potential impact is significant since the model makes predictions that can be validated experimentally. I have rarely seen a submitted manuscript that I endorse this rapidly (hence kudos to the authors), and I only have a few minor comments to make:

MAJOR

Line 211: I agree with the approach to employing bending rigidity as an opposing force for modeling purposes. That said, in the actual tip growth process, bending rigidity (although existing in pollen tubes, as covered here: Sanati Nezhad et al 2013. Lab on a Chip 13: 2599-2608 ) is likely to be irrelevant. Instead, it is the ability to for the tip to reorient its growth direction that is limiting (reorientation is covered in the following papers for example: Sanati Nezhad et al 2014. Plant Journal 80: 185-195; Bou Daher and Geitmann. 2011. Traffic 12: 1537–1551)

Therefore, I suggest that this (and potentially other) limitations of the modeling approach be spelled out somewhere.

MINOR

Line 12: Maybe add: 'female FLOWER organ'

Line 12: Why 'ALSO called'? There is only one name for the structure (stigma), hence I suggest deleting 'also'.

Line 18: Reference 3 is not related to pollen tube growth but to pollination, I believe. The role of electric fields in pollen tube guidance is covered in the following paper and older papers cited in it: Agudelo et al. 2016. Nature Scientific Reports 6: e19812

Reviewer #2: In the manuscript, entitled “Geometric and mechanical guidance: role of stigmatic epidermis in early pollen tube pathfinding in Arabidopsis”, Lucie et al. focused primarily on the mechanisms governing pollen tube elongation in stigmatic epidermal cells (papillae). They integrated computational modeling with experimental data from wild-type and katanin1-5 mutants to explore the impact of papilla morphology, pollen landing position, and guidance signals derived from mechanical cell deformation on pollen tube trajectories. This study builds upon their recent eLife paper, and the integration of computational modeling nicely demonstrates that growth guidance influenced by the mechanical properties of papilla cells is pivotal in determining pollen tube directionality during elongation.

Overall, the presented work is a beautiful example of employing computational modeling to explore the impact of morphology or mechanical properties, that are challenging to measure or manipulate experimentally, on developmental processes. The authors also discuss the potential implications for pollen tube growth in non-model species and filamentous pathogens, suggesting broad relevance to a wide audience. The presenting quality of this manuscript, including figure presentation, writing, is exceptional.

I have several comments and suggestions.

1. My first concern is about the major statement that both the geometry of the papilla and its cell wall mechanical properties guide the pollen tube elongation. In their 2020 elife paper, it is mentioned in one section title that “Shape of papilla cells is not a determining factor of the pollen tube phenotype”. It is also shown in Figure 2 that only substantial alterations in shapes, which may rarely exist in nature, affect the pollen tube trajectories. Either revising the relevant statements or investigating more on the effects of diverse papilla geometry/shape on pollen tube elongation is recommended.

2. Another concern is about how the authors addressed the effects of anisotropic cell wall mechanical properties, which is not clearly explained in the manuscript. There are several relevant points:

(1) The authors utilize the aspect ratio α to quantify elasticity differences between inner and outer cell walls. However, it is unclear how effectively this ratio is integrated into their model. Moreover, in Figure 5D, the X-axis represents cell wall stiffness rather than the aspect ratio, potentially leading to a misconception that stiffer cell walls, rather than anisotropic elasticity, are sufficient to guide longitudinal pollen tube elongation. It is strongly recommended to better elucidate which aspects of cell wall elasticity contribute to the guidance signal and pollen tube trajectory in this section.

(2) I am particularly interested in the comparison with other cell wall mutants, such as any1 mutants. While flowers of this mutant exhibit a similar papilla shape, they do not display the pollen tube coiling phenotype. It would be informative to know if these mutants have a similar aspect ratio α to knt1 mutants or wild type, or if they fall somewhere in between. Expanding the study to include another mutant could enhance the understanding of the relationship between aspect ratio α and pollen tube turning, though this is not required.

(3) There appears to be a noticeable discrepancy between the experimental data points (exp dots) and simulated lines for the knt1 mutant. It seems that anisotropic cell wall elasticity in the knt1 mutant does not entirely abolish the guidance signal, although the value is smaller than 0.01. It might be beneficial to include guidance data for the knt1 mutant in Figure 4D to provide a clearer comparison.

Other minor points:

1. It is recommended to include a table containing all major parameters and their respective biological/physical interpretations. This addition could enhance the readability for the audience.

2. It would be great if the authors could specify the software/package names or codes used for data visualization, in the Methods section.

3. In the Introduction section, paragraph 2, it is recommended to include references

4. Line 96, it is mentioned that the landing position z0 is constrained to a distance smaller than 2A. Providing context to explain this range is recommended.

5. Line 175, it is recommended to provide further clarification on what "cumulative fractions" represent upon their initial mention. This clarification would also aid in understanding Figures 3E and 4D.

6. Line 200, "as in Section", it appears that the section name is missing.

7. In the legends for Figures 3E and 4D, the statement "n = 192 pollinated papillae" needs clarification. Does it apply to all conditions (WT vs knt1, exp vs sim)?

Reviewer #3: Review of Riglet et al “Geometric and mechanical guidance: role of stigmatic epidermis in early pollen tube pathfinding in Arabidopsis”

In this manuscript, the authors investigate the role of geometrical cues and guidance force on the growth of pollen tubes on the papillae of the Arabidopsis stigma. The paper is clearly written and the authors have successfully explained the more formal aspects of their work without getting bogged down in mathematical details. Moreover, the paper touch on a very intriguing aspect of pollen tube guidance, where geometry and mechanics could replace chemical signals.

I have many comments but none of them are major. I leave it to the authors and the editor to decide which of them are worth further consideration.

1) My first reading of section “Reference trajectories of a tube on a pin-like surface” and Fig. 2 left me preoccupied that only ONE position and direction of the pollen tube were studied. It would seem difficult to draw any robust conclusion on the basis of a single set of initial conditions. Reading further, I realized that Fig. 2 is not presented as a robust result but more as a motivation of what is to come; in this case, Fig. 3 and Fig. 4 where the real results are presented. Given the confusion Fig. 2 created, perhaps the authors could consider merging Fig. 2 with Fig. 3. The pin-shaped cells could appear at the top of the current Fig. 3 to give a first illustration of the type of pollen tube behaviors that will be studied numerically.

Also, given that the analysis of Fig. 2 is very incomplete, the authors should be careful not to claim it as a result. For example, on line 155, they say: “This aligns with our previous findings on reference trajectories over a pin-like surface (Fig. 2), where we found that substantial differences in geometry are required to induce a change in the turning behaviour of an object moving along geodesics”. The claim is presumably correct but Fig. 2 falls short of establishing it beyond doubt.

2) It is not clear to me that pollen tubes in the ktn1-5 mutant follow geodesics as the authors claim (e.g. line 137). The claim is based on Fig. 3E, where observed and computed cumulative frequencies of “turns” are compared. This comparison is somewhat flawed, beginning from the fact that there are only four data points and the predicted cumulative frequency falls well outside the SEM of three of the four data points. Also, the prediction of the cumulative frequency is based on two rather strong assumptions, the pollen tube position is distributed uniformly over the papilla’s surface up to position z_0/A = 2 and the initial orientation of the pollen tube is random. While these assumptions are conservative, there are many ways in which they could be wrong for the biological system. For example, it would be reasonable to think that given a fixed patch size on the surface of the papilla, those patches closer to the pole (the top) are more likely to receive a pollen grain than those on the side. Similarly, the initial outgrowth of the pollen tube could very well prefer the principal direction on the papilla’s surface with minimal (or maximal) normal curvature. I would add to these concerns that geodesics on a surface of negative Gaussian curvature tend to be very “chaotic” because geodesics divergence on such surfaces. Therefore, small changes in initial conditions can lead to wildly different trajectories (and number of turns). Given these concerns, I think it is best not to assert strongly that pollen tubes follow geodesics.

Having said that, I don’t think the main conclusion of this paper hinges on the claim that pollen tubes follow geodesics when directional cues are missing. It would be sufficient to claim that, without guidance, (1) pollen tubes follow regular paths on the papilla’s surface and (2) many of these paths can leave the pollen tube trapped above the neck of the papillae or simply delay greatly its exit at the base of the papillae. In this context, the simulations of Fig. 2 and Fig. 3 would simply be an illustration of how complex the trajectories can be when no directional cues are present. The use of geodesics would simply be a convenient way of simulating regular paths on the papilla’s surface, without necessarily claiming that they capture the behavior real pollen tubes on ktn1-5 stigmas. Summarizing, I don’t think it is necessary to change the content of the figures, it would be sufficient to lessen the claim that pollen tubes follow geodesics.

Minor comments:

abstract “In Arabidopsis thaliana, successful fertilization relies on the precise guidance of the pollen tube as it navigates through the female {missing word} to deliver sperm cells to ovules.” I would say “female organ”.

line 40: “Our main findings showed that the coiled paths of WT pollen tubes on ktn1-5 papillae stem from the absence of guidance,” As far as I can tell, this sentence and a similar sentence in the discussion (line 291) are the only mentions that the experiments involve the growth of a WT pollen tubes on mutant papillae. It would be worth clarifying this important experimental fact in the abstract and/or the results section. My first inclination was to assume that a pollen tube with mutated katanin could have reduced “steering” ability.

line 55 and Fig. 1 G, I, J. I would suggest using variable names that are more descriptive than A, B, C, D. Although all the information is available for identifying the correspondence between the variables and the papilla geometry, the nondescript names are more cumbersome than using, for example, D_tip, D_neck, W_tip, W_neck. The variable “A” is especially important to understand Figs. 3. and 4. It deserves a more informative name.

line 68: “Geodesics correspond to the length-minimising curves between pairs of points and possess the remarkable property that, at any given point on the surface and in any specified direction, a single, unique geodesic exists [8]”

line 74: Can 38% be justified. It seems random. Also, it would be good to clarify the number 0.43 and 0.27 which are confusing because they are nondimensionalized.

A few comments about Fig. 3 and 4:

(i) The black arrows to indicate the position of the pollen grain is useful but it seems an arrow initiating at the start point and pointing in the direction of growth would be easier to interprete.

(ii) The phase diagram shows z_0/A (e.g. Fig. 3 D) while the parameters values are given as z_0 = 7.5 µm ( Fig. 3C and elsewhere). Although the simulations are located in the phase diagram, I would be useful to report z_0 / A.

(iii) Labeling in the figures tends to be too small, especially in Figs. 3 and 4. The reader is forced to zoom in and out to read the main text and then study the figures.

line 150: “To integrate the entire range of configurations we simulated, we represented …” eliminate “we simulated”

line 170: “At the opposite, when a” I would say “In contrast”

line 197: “deviation from geodesic trajectories requires the presence of a lateral force in the tangent plane on the papilla surface.” I think the word “lateral” in this sentence is confusing. Lateral to what? I believe the statement is clearer without the word “lateral”, i.e. “… the presence of a force in the tangent plane …”; or you may want to say “… the presence of a righting force in the tangent plane …” (see below)

line 199: “We therefore performed numerical simulations of pollen tube trajectories using the same framework as in section (see also Methods)” The section’s name is missing.

line 201: “This alignment force does not have any preferential orientation towards the papilla pole or base and is proportional to sin (2ϕ)”. This should be rephrased. I think you are trying to say that the righting force is symmetrical with respect to ϕ = 90˚ and does not “favor” the base over the pole but it is true that the axis of the papilla (towards the tip or the base) is the preferred orientation. Also, the most intuitive interpretation of the force is to imagine the vector to reach its maxima when pointing downward (ϕ= 0˚) and upward (ϕ = 180˚). So, naively, we would expect the force to be proportional to cos (2ϕ). I think it would help the reader if you stated that it is a RIGHTING force whose magnitude increases as the tube deviates from the axis of the papilla.

line 206: “For growth oriented towards the papilla pole (90◦ ≤ ϕ < 180◦), the growth direction will align completely with the long axis towards the papilla base.” Do you mean “… will align completely with the long axis towards the papilla POLE” ?

line 208: “The magnitude of this alignment force is determined by the adimensional constant μ,” It seems the magnitude also depends on the orientation of the pollen tube. Perhaps you could say, “The maximal value of this alignment force is set by …”

line 418 “Eq. (12) can be derived from a variational principal” Should be “variational principle”

Reviewer #4: The authors present an elegant mechanical model to explain how pollen tubes may be guided at the stigma, from the tips of elongated papilla cells, where the pollen grains germinate, to their bases. With their model, they claim that guidance may proceed by mechanical cues alone. The model is based on their previous observations (mainly Riglet et al. 2020, eLife) that 1) pollen tubes on these cells grow inbetween two layers of the cell wall and 2) the guidance mechanism does not function in the katanin mutant, which has a softer cell wall (lower Young's modulus) and an outer cell wall layer that is relatively more extended by the pollen tube than the inner layer compared to wild type.

This mechanical guidance is interesting, as it differs from the more common type of guidance by chemical cue lower in the style.

Overall, the manuscript is well written.

Major comments:

* My major question throughout is: is it really the difference in stiffness of the different layers that underpins the mechanical guidance, or is the reduced anisotropy of the cell wall the bigger issue? This is important to address, as the katanin mutant is know for reduced anisotropy in the cortical microtubule array and, consequently, of cellulose microfibril layers. In their earlier work, the authors find that oryzalin treatment (a microtubule depolymerizing drug) increases the number of turns taken by the pollen tube, though not as dramatically as the ktn1-5 mutation. They also find that various other cell wall mutants do not have a pollen guidance phenotype (I quote from Riglet et al 2020: "We selected mutants impaired in the cellulose synthase complex (kor1.1, prc1 and any1), hemicellulose biosynthesis (xxt1 xxt2, xyl1.4) and pectin content (qua2.1), for which expression of the corresponding genes in stigma was confirmed (Figure 5—source data 2). Strikingly, none of the 6 cell wall mutants displayed the coiled pollen tube phenotype (Figure 5—figure supplement 3A–F)."). In their quantification of the papilla cell wall Young's modulus, there was no significant difference between ktn1-5 and xxt1/xxt2.

While I understand that anisotropic mechanical models can be substantially harder than isotropic ones, the option that anisotropy is the key factor should at the very least be thoroughly discussed, including the experimental evidence for either scenario.

* The model where the wall is extending only looks at the equilibrium situation on the shaft of the pollen tube, not on the growing tip where the actual "decisions" are made. Again, I see the difference in challenge level. When looking at this aspect, however, perhaps an anisotropic model is feasible? An important point to think about (possibly just in the discussion) is how local/global the effects of cell wall deformation by the pollen tube are. Figure 5 could nicely illustrate this discussion.

* In the introduction and throughout the text and figure legends, I find it often hard to distinguish between what are new observations/analyses and what was already known from Riglet et al 2020 and Reglet et al 2021 Plant signaling & behaviour. This should be improved.

* In the discussion, I miss certain points:

- the observations in the two previous Riglet papers, particularly where relating to other (cell wall related) mutants

- what is known about the molecular basis of a persistence of pollen tube growth direction and how would that constrain the relevant parameter range? Passive guidance or active reorientation of the growth direction in response to perceived cues? (c.f., the curling of Medicago truncatula root hairs in response to NOD factors depends on a ROP and involves quite high curvature: Lei et al, 2015, plant cell, "The small GTPase ROP10 of Medicago truncatula is required for both tip growth of root hairs and nod factor-induced root hair deformation")

Importantly: does this allow for approximating the pollen tube as an "orthotropic inflatable beam" (REF 4 in the supplementary, source of eqn 19).

Minor comments:

The legend of figure 1 does not indicate which panels merely repeat information from Riglet et al 2020 (with different but similar pictures) and which show actually new information (G-J only, I think).

As half of this figure serves to help the reader understand the context anyway, I think it would be good to also show the equivalent of 1D for the ktn1-5 mutant.

Figure 5B,C: with the chosen font/font size, r_i looks like the symbol \eta, which makes the legend/text much harder to understand.

Line 357: which tools in ImageJ were used, for which figures/tables?

Line 379/380: "To integrate these equations we used the odeint function of the Python SciPy library [22]." What is the reason for not sharing the code for these calculations?

Supplementary material: please repeat the schematics that introduce the relevant parameters for easier reading.

Further schematics explaining the derivation would also be helpful.

"We assume that the pollen tube is separating and softening the inner and outer papilla cell layers over a width of 2r" -- please justify this choice and/or explore a wider range of effects.

Typos etc:

main text equation 1: use \cdot before "W"

line 365: please use [0,2\pi) for the interval

main text eqn 10: symbol R' is introduced half a page later. This is annoying.

figure 1 legend: a latex error just before label "J:". Change "<" into "$<$"

Also: I'd say the depicted stigmas have more than (a few) dozens of papillae.

"a elastic modulus" -> "an elastic modulus"

**Have the authors made all data and (if applicable) computational code underlying the findings in their manuscript fully available?**

Reviewer #1: None

Reviewer #2: Yes

Reviewer #3: Yes

Reviewer #4: **No: **All measurement data is available in tables, but I did not see any code for the numerical calculations. As the manuscript does not describe methodological advances/new software, this is a "grey zone" case.

PLOS authors have the option to publish the peer review history of their article (what does this mean?). If published, this will include your full peer review and any attached files.

Reviewer #1: No

Reviewer #2: No

Reviewer #3: **Yes: **Jacques Dumais

Reviewer #4: No
---

## [Decision Letter · Decision Letter 1]

19 Dec 2024

PCOMPBIOL-D-24-00855R1

Geometric and mechanical guidance: role of stigmatic

epidermis in early pollen tube pathfinding in Arabidopsis

PLOS Computational Biology

Dear Dr. Fobis-Loisy,

Thank you for submitting your manuscript to PLOS Computational Biology. After careful consideration, we feel that it has merit but does not fully meet PLOS Computational Biology's publication criteria as it currently stands. Therefore, we invite you to submit a revised version of the manuscript that addresses the points raised during the review process.

I am again asking for a major revision to give you an opportunity to incorporate justified comments of two of the three reviewers. In particular, as pointed out by reviewer #3 and #4 several conclusions needs to be nuanced and alternative explanations, including anisotropy, must be clearly considered. 

Please submit your revised manuscript within 60 days Feb 18 2025 11:59PM. If you will need more time than this to complete your revisions, please reply to this message or contact the journal office at ploscompbiol@plos.org. Please include the following items when submitting your revised manuscript:

We look forward to receiving your revised manuscript.

Kind regards,

Roeland M.H. Merks, Ph.D

Academic Editor

PLOS Computational Biology

Jason Haugh

Section Editor

PLOS Computational Biology

**Journal Requirements:**

Please ensure that the funders and grant numbers match between the Financial Disclosure field and the Funding Information tab in your submission form. Note that the funders must be provided in the same order in both places as well.

**Reviewers' comments:**

Reviewer's Responses to Questions

**Comments to the Authors:**

Reviewer #2: I appreciate the addition of SEM images for the cell wall mutant, which enhances the clarity of the findings. The estimation of the aspect ratio ( ) for any1 and xxt1xxt2 papillae being equal to 1 (similar to WT) seems reasonable to me. I also value the inclusion of the new supplemental table (Table S1), which presents the major parameters in their model. Overall, I do recommend the publication of this manuscript.

Reviewer #3: I thank the authors for the careful revision of their manuscript. They answered my comments to my complete satisfaction. I only have a few minor comments left.

- The Clairaut’s theorem is very useful to clarify the interpretation of the geodesics. Thank you for adding it.

- line 247: “In summary, our simulations show that in the absence of guiding cue from the stigma side, the pollen tube trajectory is entirely determined by its initial position z0 and initial direction ϕ0 (modulated by the self-avoidance property).”

I am baffled by this statement. It seems the authors are claiming as a result (i.e. “our simulations show”) what was in fact THE IMPLEMENTATION of the simulations. Concretely, it is the authors themselves who created simulations where the trajectory of the pollen tube follows the geodesic specified by the initial position and orientation of the pollen tube. It is not surprising that the simulations “show” this property. I would rephrase this summary statement.

line 419: “Overall, we demonstrated here that the cylindrical shape of the papilla and the cell wall elasticity synergistically exert a torque on the advancing pollen tube tip, acting as a guidance cue along the longitudinal papilla axis.”

Again, the use of the word “demonstrated” is not appropriate here. It is the authors who postulate that the shape of the surface and the wall elasticity are guiding the growth of pollen tube. There is little experimental evidence to support the claim, and it is certainly not demonstrated either empirically or numerically. Please, present this as a hypothesis for the guidance of pollen tubes that seems roughly compatible with the limited observation available on the wall deformation/mechanical properties.

Reviewer #4: See attached file

**Have the authors made all data and (if applicable) computational code underlying the findings in their manuscript fully available?**

Reviewer #2: Yes

Reviewer #3: Yes

Reviewer #4: Yes

PLOS authors have the option to publish the peer review history of their article (what does this mean?). If published, this will include your full peer review and any attached files.

Reviewer #2: No

Reviewer #3: **Yes: **Jacques Dumais

Reviewer #4: No

**Figure resubmission:**
---

## [Editor Report · Decision Letter 2]

21 Apr 2025

Dear Dr Fobis-Loisy,

We are pleased to inform you that your manuscript 'Geometric and mechanical guidance: role of stigmatic epidermis in early pollen tube pathfinding in Arabidopsis' has been provisionally accepted for publication in PLOS Computational Biology.

Best regards,

Roeland M.H. Merks, Ph.D

Academic Editor

PLOS Computational Biology

Feilim Mac Gabhann

Editor-in-Chief

PLOS Computational Biology

---

## [Editor Report · Acceptance letter]

PCOMPBIOL-D-24-00855R2

Geometric and mechanical guidance: role of stigmatic

epidermis in early pollen tube pathfinding in Arabidopsis

Dear Dr Fobis-Loisy,

I am pleased to inform you that your manuscript has been formally accepted for publication in PLOS Computational Biology. Your manuscript is now with our production department and you will be notified of the publication date in due course.

With kind regards,

Anita Estes
